# Injectable bioresorbable conductive hydrogels for multimodal brain tumor electroimmunotherapy

Amit Singh Yadav [1], Umut Aydemir [1], Karin Hellman [1], Peter Ekström [1], Abdelrazek H. Mousa [2,3], Jiaxin Li [4,5], Muhammad Anwar Shameem [2], Cedric Dicko [6], Johan Bengzon [4,7], Fredrik Ek [1], Martin Hjort [1] & Roger Olsson [1,2,8] ✉

Current electrode technologies are too rigid for safe and effective delivery of electrotherapy in the brain, and patients with glioblastoma continue to face a devastating prognosis, with median survival stalled at 15 months despite intensive treatment with surgery, radiation, and chemotherapy. But these conventional approaches potentially compromise immune function, underscoring the urgent need for therapies that activate, rather than suppress, the immune system. Therefore, we introduce injectable conductive hydrogels engineered to match the softness of brain tissue while exhibiting electrical conductivities up to three orders of magnitude higher than any previously reported injectable hydrogels. They can be implanted through minimally invasive syringe capillaries as narrow as 30 μm—avoiding brain tissue damage—and via convection-enhanced delivery (CED) or endovascular catheters, the latter potentially eliminating the need for open brain surgery. Additionally, it can drape a resection cavity to eliminate residual tumor cells. In human glioblastoma tumors in the chicken chorioallantoic membrane model, implantation of the electrode using CED, followed by irreversible electroporation, obliterated tumors within three days. Other injection techniques impaired tumor growth, induced immunogenic cell death, and a robust infiltration of helper and cytotoxic T cells, alongside macrophages, highlighting the immune-activating and tumor-targeting capabilities.

The prognostic outcome for glioblastoma (GBM) patients has been stagnant for decades. The median overall survival in well-selected patients in clinical trials following diagnosis remains 15 months after the conventional therapeutic strategy of surgery, radiotherapy, and chemotherapy. GBM is a highly invasive primary brain cancer that is characterized by significant inter- and intra-tumoral diversity, contributing to its resistance to currently available treatments[1]. Therefore, GBM should be an extraordinary target for the immune system that recognizes a tremendous diversity of antigens. However, while the new cancer immunotherapies (e.g., checkpoint inhibitors) have had a significant impact on other cancers (e.g., lung, melanoma, and breast), they have so far had limited success for GBM. This limited

[1]Department of Experimental Medical Science, Chemical Biology & Therapeutics, Lund University, Lund, Sweden. [2]Department of Chemistry and Molecular Biology, University of Gothenburg, Gothenburg, Sweden. [3]Chemistry Department, King Fahd University of Petroleum and Minerals, Dhahran, Saudi Arabia. [4]Neurosurgery, Department of Clinical Sciences Lund, Faculty of Medicine, Lund University, Lund, Sweden. [5]Lund Stem Cell Center, Lund University, Lund, Sweden. [6]Pure and Applied Biochemistry, Department of Chemistry, Lund University, Lund, Sweden. [7]Kamprad Laboratory, Department of Clinical Sciences, Lund University, Lund, Sweden. [8]Department of Chemistry, Lund University, Lund, Sweden. ✉e-mail: roger.olsson@med.lu.se

response is intriguing given recent findings that tumor-targeting T cells are present in the skull bone near the tumor site in GBM patients[2]. Although combining surgery with radiotherapy and chemotherapy is the most effective approach for rapidly eliminating most cancer cells in the short term, it might compromise the immune response, which is crucial for addressing residual cancer cells[3]. Thus, chemotherapies and radiotherapy have shown limited effect. An alternative or adjuvant treatment might be electrotherapy, which can ablate the primary tumor and induce an anti-tumor immune response. Electrotherapy is an emerging approach to non-thermally ablate solid tumors. Irreversible electroporation (IRE) uses high-voltage pulsed electric fields to disrupt cell membranes, creating nanoscale pores in cancer cells. Notably, IRE has been demonstrated to induce a systemic anti-tumor immune response by triggering immunogenic cell death (ICD)[4]. Although electrotherapies are clinically used in several countries, their widespread application is often limited by the reliance on multiple rigid electrodes and a high degree of invasiveness. These techniques have reduced efficacy for large or irregular tumors due to their limited ability to eliminate the tumor cells completely, as effectiveness decreases with distance from the electrode. This has led to stringent inclusion criteria for treating patients. Moreover, using several rigid electrodes restricts their applicability in sensitive tissues, such as brain tissue. Consequently, current electrotherapies are not used for treating brain tumors, large or irregular solid tumors, or tumors located near vital organs.

A solution to this problem is bioresorbable in situ assembled injectable non-substrate-bound conductive hydrogels as electrodes to match brain softness and avoid the insertion and removal of the electrodes using surgery. Various reports have disclosed injections of premade hydrogels or in vivo assembly of conductive polymers by electrical or enzymatic polymerization within tissues[5–8]. A recent evaluation of injectable conductive hydrogels revealed low conductivities of the resulting electrodes, comparable to those of biological tissues and fluids, in the best cases, such as blood ($10-20\ mS\ cm^{-1}$)[8].

Designing minimally invasive injectable hydrogels with sufficient conductivity to enable IRE remains a significant challenge. In our previous work, we reported the discovery of a nanoparticle mixture comprising A5, a derivative of the self-doped, water-soluble, mixed ion−electron conducting polymer poly(3,4-ethylenedioxythiophene)-butoxy-1-sulfonate (PEDOT-S), in combination with 2-(2,5-bis(2,3-dihydrothieno[3,4-b][1,4]dioxin-5-yl)thiophen-3-yl)ethyl(3-(trimethylammonium)propyl) phosphonate (ETE-PC) (Fig. 1a)[9–11]. This mixture self-assembles in tissues (e.g., brain, heart, and muscles) to form conductive hydrogel structures. The mixture was implanted in tissue minimally invasively by thin capillaries the size of a human hair ($30\ \mu m$), a size reported not to significantly damage brain tissue−by basically displacing cells and not disrupt them−we verified this by injecting A5:ETE-PC into brains, a minor inflammatory response was seen that resolved within 2 days. Further, the A5:ETE-PC hydrogel demonstrates excellent biocompatibility, brain-compatible Young's modulus ($2-27\ kPa$), and is bioresorbed, avoiding removal by surgery[10,11].

In this study, we demonstrate the in situ assembly of highly conductive hydrogels within tumors to enable IRE-mediated tumor ablation and immune activation as a novel electroimmunotherapy approach for glioblastoma. We demonstrate that the electrode can be implanted using three distinct approaches: (i) via injection with capillaries, including convection-enhanced delivery, (ii) through the vascular system using endovascular catheters, and (iii) placed as a drape in the resection cavity after surgery. We use the chicken chorioallantoic membrane (CAM) in vivo model for glioblastoma. The CAM model uniquely integrates vascular, immunological, and geometric features essential for evaluating electro-immunotherapeutic strategies that are difficult to realize in either current three-dimensional organoid systems or orthotopic rodent models. Its vascularized extra-

embryonic membrane supports the rapid growth of human tumors to 5–10 mm in diameter within 3–7 days, dimensions realistic to enable the implantation of electrodes. Additionally, the CAM harbors an immune system that can be augmented with transplanted human immune cells, enabling real-time assessment of electro-induced immunomodulation. In contrast, canonical cancer organoids typically remain 0.1–0.6 mm in size and develop necrotic cores once their radius exceeds approximately 0.5 mm, limiting their utility for electrode placement and obscuring vascular or immune endpoints. Orthotopic rodent glioblastoma models, while preserving brain architecture, necessitate euthanasia when intracranial tumors reach 4−6 mm due to neurological morbidity. Collectively, the CAM model offers millimeter-scale tumor volumes, rapid vascularization, manipulable immune contexture, and unrestricted surgical access. In addition, the CAM model has demonstrated exceptional translational value as a patient-derived xenograft model for high-grade gliomas. In a previous study, the CAM xenograft response mirrored the clinical outcomes of fourteen patients with a complete (100%) correlation[12].

IRE experiments were conducted in the chicken CAM model by implanting 3D electrode assemblies using different injection techniques. Specifically, the A5:ETE-PC mixture was injected into the tumor by column-injections, a technique used in cell transplantation, through capillaries down to $30\ \mu m$ in diameter or by convection-enhanced delivery (CED). The resulting electrode structure is bioresorbable, biocompatible, and has an unprecedented hydrogel in vivo conductivity of $6\ S\ cm^{-1}$ within the tumor−three orders of magnitude higher than previously reported values (Fig. 1h), enabling its use for high-voltage applications. Moreover, IRE treatment using the A5:ETE-PC formed hydrogel electrode from column-injections effectively inhibited tumor growth and induced immunogenic cell death, accompanied by increased immune cell infiltration, highlighting its potential for synergistic tumor ablation and immune modulation. Thus, electrotherapy resulted in a two- to three-fold increase in helper and cytotoxic T cells, as well as a robust four-fold infiltration of macrophages in the tumor. However, using convection-enhanced delivery, an electrode network was formed throughout the entire tumor. After applying IRE, the tumors were obliterated within a day. This highlights its immune-activating potential and ability to target and eliminate tumor cells directly.

## Results and discussion

### Preparation and characterization of A5:ETE-PC 3D flexible electrode

We have previously reported that electrofunctionalization of A5 with ETE derivatives adds modularity to its structure[10]. In this study, we used ETE-PC to functionalize A5 (Fig. 1a) to generate 3D flexible branched electrodes within the tumor. ETE-PC has a low oxidation potential (0.47 V vs. Ag/AgCl) and polymerizes below the oxidation potential of water, making it safe for electrofunctionalization within the tissue. UV−visible spectra of ETE-PC demonstrate a sharp peak of unpolymerized ETE-PC around 350 nm, while polymerized ETE-PC shows a broad absorption in the range of 600−900 nm with a maximum around 780 nm (Supplementary Fig. 1a). ETE trimers exhibit fluorescence with a maximum excitation peak at around 375 nm. ETE-PC displayed emission in the range of 410−500 nm, with an emission maximum at 435 nm (Supplementary Fig. 1b). The injection of A5:ETE-PC ($20{:}40\ mg\ ml^{-1}$) nanoparticles into a 0.5% agarose gel leads to aggregation of A5 in the hydrogel by ionic crosslinking. At the same time, ETE-PC diffuses into the agarose matrix out from the hydrogel. Diffused ETE-PC was electrofunctionalized onto the A5 by applying a bias through the A5 self-assembled electrode; this resulted in the formation of a branched 3D flexible electrode. This structure consists of an A5 core intercalated with polymerized ETE-PC with ETE-PC dendrites protruding from the A5:ETE-PC core in all directions (Supplementary Fig. 2a−c). ETE-PC concentration offered the possibility to

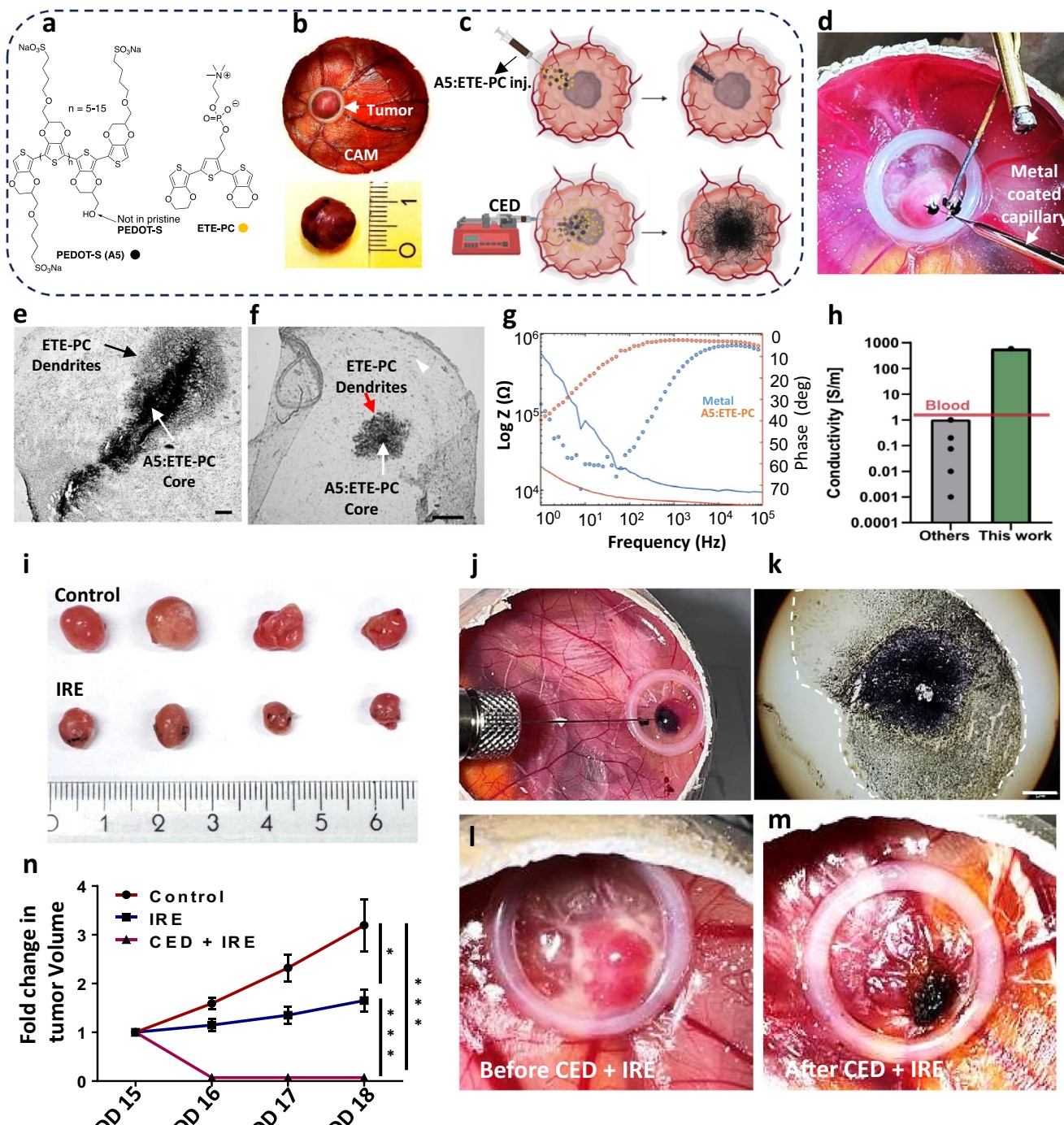

**Fig. 1 | Intra-tumor assembly and electrotherapy via A5:ETE-PC conductive hydrogel electrode. a** Chemical structures of A5 and ETE-PC. **b** Representative images of U87 tumors developed in CAM model. **c** Schematic representation of A5:ETE-PC electrode assembly via capillary column-injection or convection-enhanced delivery (CED). Created in BioRender. Yadav, A. (2025) (https://BioRender.com/8864ndt). **d** Microinjection and electrofunctionalization of A5:ETE-PC in the tumor. **e, f** Longitudinal section and transverse sections of A5:ETE-PC electrode showing A5:ETE-PC core (white arrows) and ETE-PC dendrite (red arrows) formation in tumor; scale bar **e**: 100 μm, scale bar **f**: 500 μm. **g** Impedance analysis in the tumor using metal (Pt-Pd coated capillary) or A5:ETE-PC electrode. **h** Comparison of the electrical conductivity of the A5:ETE-PC in tissue compared to previously reported injectable hydrogels. **i** Digital photographs of IRE-treated tumors harvested on EDD18. **j** Convection-enhanced delivery (CED) of A5:ETE-PC in U87 tumor using a Hamilton syringe attached to a syringe pump. **k** U87 tumor cryosection (20 μm) showing extended A5:ETE-PC electrode assembly after CED. **l, m** Images demonstrating ablation of U87 tumor before and after CED of A5:ETE-PC and subsequent IRE treatment. **n** Tumor volumes from A5:ETE-PC mediated IRE-treated tumors injected via capillary or CED were calculated along with controls and analyzed. The graph represents the fold change in tumor volume to initial tumor size, mean ± SE, n = 11 for control and IRE groups, n = 6 for CED + IRE group. Student's *t* tests using the Holm–Šídák correction for multiple comparisons were used (without any adjustment) to perform two-sided analysis of variance in the fold change in tumor volume on each measurement day; *p = 0.016 for Ctrl vs IRE, ***p = 0.00076 for Ctrl vs CED + IRE, ***p = 0.00017 for IRE vs CED + IRE.

tune the dendrite geometry, as well-defined dendrites were observed above 40 mg ml$^{-1}$ (Supplementary Fig. 2d). In our previous study, we demonstrated that core A5 forms hydrogels with a water content of >90%[9]. The water content of electrofunctionalized A5:ETE-PC was evaluated here, which resulted in a water content that is more than 20 times the polymer weight (Supplementary Fig. 2e).

The electrofunctionalization process of ETE-PC on A5 was further analyzed using cyclic voltammetry (CV). The A5:ETE-PC solution was injected into an agarose mold, and CV was performed. The cyclic voltammogram exhibited three oxidation peaks at 0.1, 0.47, and 0.65 V, respectively (Supplementary Fig. 3a). Gerasimov et al. have shown that ETE-PC undergoes oxidative polymerization at 0.47 V, indicating that the middle peak represents electrofunctionalization of ETE-PC on A5[13]. We further compared the CV (0–1 V, at a scan rate of 0.05 V s$^{-1}$) of the A5 with the A5:ETE-PC electrode after electrofunctionalization. The absence of the peak associated with ETE-PC oxidative polymerization indicates its nearly complete electrofunctionalization onto A5. Furthermore, the spectra revealed an increased area between 0.1 and 0.8 V, indicative of higher capacitance following the electrofunctionalization of ETE-PC on A5 (Supplementary Fig. 3b). This enhanced capacitive property is attributed to the larger dimensions of the A5:ETE-PC electrodes compared to A5 alone, resulting in a greater effective surface area for charge storage. To further investigate how ETE-PC electrofunctionalization affects the electrochemical properties of the A5 electrode, we conducted electrochemical impedance spectroscopy (EIS) using three-electrode setup (Supplementary Fig. 4a). Results showed that the A5:ETE-PC hydrogel electrode has a significantly lower impedance than A5, emphasizing its improved conductive properties (Supplementary Fig. 4b). The Nyquist plot suggests that charge transfer resistance and double layer capacitance contribute to impedance in both A5 and A5-ETE-PC electrodes; however, both real and imaginary resistance were significantly lower for A5:ETE-PC electrode (Supplementary Fig. 4c). The dendritic structures in the A5:ETE-PC electrode provide an enhanced electrode–tissue interface for charge transfer. Further, the lower phase angle indicates less capacitive reactance of the A5:ETE-PC electrode due to higher double-layer capacitance correlating with the CV results (Supplementary Fig. 4d). Higher capacitance leads to lower imaginary resistance in electrofunctionalized electrodes as compared to A5 alone. Moreover, EIS data also suggests that the A5:ETE-PC electrode provides a linear and stable system (Supplementary Fig. 4e). Overall, the data indicate that the A5:ETE-PC electrode forms a stable electrochemical system with low impedance.

Recent publications have shown bioresorption of PEDOT derivatives by bioerosion[14–16]. Therefore, we performed in vitro experiments in which we prepared an A5:ETE-PC electrode on an agarose gel and kept it in PBS (pH 7.4) at 37 °C for 2 weeks. Electrofunctionalized A5:ETE-PC was found to erode over time, verifying bioerosion as a plausible clearance pathway. The extent of bioerosion was tunable by varying the electrofunctionalization duration, providing a practical and straightforward lever to control bioresorption kinetics (Supplementary Fig. 5). Consistent with Montazerian et al.[16], who observed immune-cell accumulation around their hydrogel, we noted a similar response following irreversible electroporation (IRE), suggesting an additional elimination route via phagocytic clearance. In line with this, we observed that macrophages readily phagocytosed A5:ETE-PC in vitro (Supplementary Fig. 6).

## Chicken chorioallantoic membrane (CAM) tumor model of glioblastoma

To validate our in vitro findings in an in vivo setting, we used the chicken chorioallantoic membrane (CAM) tumor model. The CAM model has been widely reported as an effective in vivo system for studying tumor progression with proven translational value as a PDX model for high-grade gliomas[12]. It provides key stromal factors essential for establishing a reliable tumor microenvironment, including immune cells, extracellular matrix components, and vascular and lymphatic networks. Additionally, the CAM is a validated model for preclinical cancer research, particularly for evaluating angiogenesis, tumor growth, metastasis, immunological studies, and therapy responses[17,18]. Fertilized chicken (*Gallus gallus*) eggs were placed in a specialized egg incubator at 37.5 °C with 65% relative humidity. The day the eggs were placed in the incubator was designated embryonic development day 0 (EDD0). Xenografts were developed by implanting glioblastoma cells on CAM. The average tumor size ranged from 5 mm, with growth reaching 1 cm by EDD18 (Fig. 1b and Supplementary Fig. 7). The tumors were highly vascularized and large enough to facilitate electrode implantation and subsequent studies.

## Intra-tumor electrode assembly and irreversible electroporation

Intra-tumor electrode assembly was achieved by two approaches utilizing the fluidic nature of the electrode solution; injection via thin capillary using a column injection technique used for cell transplantation or by convection-enhanced delivery using syringe pump (Fig. 1c). In the first approach, a 30 μm metal-coated capillary was used to microinject and electrofunctionalize the A5:ETE-PC mixture (40 mg ml$^{-1}$ A5, 20 mg ml$^{-1}$ ETE-PC) (Fig. 1d). During electrofunctionalization, oxidative currents were observed, indicating effective functionalization of ETE-PC on the A5 scaffold (Supplementary Fig. 8a, b). EIS investigations assessed the electrochemical properties of A5:ETE-PC electrodes compared to coated metal electrodes in tumor tissue. Following electrofunctionalization, tumors were harvested and cryosectioned, and electrical measurements of the A5:ETE-PC electrode were conducted in the tumor section on interdigitated Au electrodes.

Microscopic analysis of tumor sections revealed ETE-PC dendrites extending radially from the A5:ETE-PC core in all directions with up to 0.5 mm diameter, confirming the successful in situ assembly of a 3D electrode structure within the tumor (Fig. 1e,f). The diameter of typical needle electrodes for effective IRE ranges between 0.4 and 1.5 mm. 3D reconstruction of injected electrodes in the tumor, post-tissue clearing procedure, also demonstrated 3D structure with continuous dendrites extending radially (Supplementary Fig. 9).

The EIS results showed the A5:ETE-PC electrodes provide a stable electrical system with significantly lower impedance within the tumor tissue as compared to metal electrodes, indicating the higher conductive property of the electrode in the tumor microenvironment (Fig. 1g and Supplementary Fig. 10a, b). We also performed impedance analysis before and after electrofunctionalization, i.e., before and after dendrite formation. The results show that the A5:ETE-PC before electrofunctionalization demonstrates a decrease in impedance, which is further decreased significantly after electrofunctionalization as compared to the metal electrode (Supplementary Fig. 10c). Electrical measurements of the A5:ETE-PC electrode within tumor tissue demonstrated a high conductivity of 6 S cm$^{-1}$, three orders of magnitude higher than that of injectable conductive hydrogels reported previously (Fig. 1h and Supplementary Fig. 11a–d)[8,15,16,19,20]. It can be attributed to the enhanced electrode-tissue interface area significantly reducing charge transfer resistance and enhancing the double layer capacitance, leading to low capacitive reactance. Moreover, since both A5 and ETE-PC are intrinsically conductive, their hydrogel electrode is conductive throughout the entire network—unlike other hydrogels that generally are based on nonconductive polymer matrices supplemented with conductive fillers. This intrinsic conductivity explains why A5:ETE-PC electrodes exhibit higher conductivity than conventional hydrogels. Overall, it substantially reduces the impedance, leading to higher conductivity in tissue.

We further investigated the therapeutic efficacy of A5:ETE-PC hydrogel in cancer electrotherapy via IRE. In situ assembled electrodes

were used to apply multiple short-duration high-voltage pulses on EDD15, and tumor growth was measured until EDD18, after which tumors were excised, imaged, weighed, and processed for further analysis. Visual observation of the excised tumors revealed noticeably smaller tumor sizes in the IRE-treated group compared to the control group (Fig. 1i). In a separate experiment, IRE-treated tumors were cryosectioned and subjected to immunostaining to assess Ki67 expression as a proliferation marker. The results demonstrated a significant reduction in Ki67-expressing nuclei in regions surrounding the electrode, suggesting a marked inhibition of tumor cell proliferation (Supplementary Fig. 12a). Further, multiple overlapping images of the tumor sections were captured and stitched to generate a composite image of the entire section. Ki67-positive (Ki67+) nuclear staining was then analyzed. Composite binary images of the tumor sections revealed noticeably lower numbers of Ki67+ nuclei in the IRE-treated sections compared to the control group (Supplementary Fig. 12b). The area and number of Ki67+ nuclei were measured from each group to quantify these differences further. The proliferation index, calculated as Ki67+ nuclei/mm², was found to be higher in control tumors as compared to IRE-treated tumors (Supplementary Fig. 12c). This decrease in proliferation following IRE treatment with the A5:ETE-PC electrode aligns with the tumor growth suppression observed in our earlier analysis.

## Convection-enhanced delivery of A5:ETE-PC and tumor ablation

We further executed CED to extend the volumetric coverage of the A5:ETE-PC hydrogel electrode within the tumor. CED is a clinically used method to locally deliver therapeutic agents into brain tumors using catheters connected to a pump that provides a continuous, positive-pressure micro-infusion of desired agents. However, this technique has not been reported to be used for the implantation of electrodes previously. To emulate CED, the A5:ETE-PC mixture was infused at a slow rate (10 μl/h, a clinically adapted rate) in U87 tumors using a thin needle (108 μm) Hamilton syringe connected to a syringe pump (Supplementary Fig. 13a). Post-infusion, the material was electro-functionalized, and IRE was applied. The result demonstrated a clear change in tumor color from pink to black as the infusion progressed, indicating an enhanced distribution of A5:ETE-PC in the tumor (Fig. 1j). Further, cryosection of tumor showed a dark network like structure extending most of the tumor demonstrated that CED enhances the distribution of A5:ETE-PC (Fig. 1k). A5:ETE-PC electrode assembly post-CED significantly lowered the impedance in the tumor as compared to the metal electrode, making it amenable for IRE (Supplementary Fig. 13b). Most interestingly, IRE post-CED led to complete obliteration of tumors in most cases (Fig. 1l, m and Supplementary Fig. 14a). Further, quantitative analysis of the growth of column-injected tumors demonstrated a prominent decrease in mean tumor volume in the IRE-treated group observed over 4 days relative to the control group (Fig. 1n and Supplementary Fig. 14b, c). The effect of the CED group was more dramatic with significant obliteration of tumors within 24–48 h (Fig. 1n). In addition, the CED-IRE treatment left a drape behind, which potentially can be further used on recurrent glioblastoma. These results were further consolidated by tumor weight measurements, where a significantly lower tumor weight was observed in the IRE-treated group; it was not possible to analyze the weight of the CED-IRE group (Supplementary Fig. 14d). These results demonstrate the astonishing effectiveness of the CED-IRE combination in the ablation of glioblastoma in vivo.

## Drape formation

GBM treatment remains challenging even after surgery due to residual cells present deep within the margins of GBM resections, leading to a high recurrence rate[21]. To target these cells, we designed a novel electrotherapy approach that involves covering the resection cavity margin with a drape comprised of A5:ETE-PC with ETE-PC dendrites

extending deep into the GBM resection margin (Fig. 2a, e). Surgical resection of the GBM tumor on the CAM resulted in forming a resection cavity (Fig. 2b, c). The subsequent procedure was designed with translational applicability in mind: the cavity was filled with the A5:ETE-PC nanoparticle mixture, and after a 15-min incubation, the excess material was extracted. Electrofunctionalization then produced a uniformly distributed black drape that conformed to the cavity surface and dendrites protruding into the resection margin (Fig. 2d, e). Further, EIS analysis revealed more than a 3-fold decrease in impedance when the cavity was covered with a drape compared to without a drape (Supplementary Fig. 15a). Applying modest electrical biases at varying distances over the drape resulted in distance-dependent currents in the microampere (μA) range. Even when the electrodes were kept around 4 mm apart (almost the diameter of the cavity), we found a current of 1 μA, showing that the drape is highly conductive (Supplementary Fig. 15b, c).

We have previously demonstrated that the polymer's mechanical modulus in agarose and zebrafish brains and hearts adapts to the surrounding material[10,11]. Similarly, both cavities, with and without polymer-coated have low Young's moduli of 1–4 kPa. This mechanical matching enables the tissue and polymer to respond similarly to external stresses (Supplementary Fig. 16).

To simulate the therapeutic effect of drape, we perform IRE experiments with cancer cells, using an in vitro model with the A5:ETE-PC hydrogel electrode interfaced with U87 cells in agarose (Supplementary Fig. 17a). Cells were stained with Calcein-AM to facilitate visualization. Microscopic analysis demonstrated dendritic structures branching from A5:ETE-PC layer covering the agarose wall and surrounding the cells (green fluorescence) (Fig. 2f, g). No adverse changes in cell morphology were observed, indicating the biocompatibility of the electrodes.

EIS studies exhibited significantly reduced impedance compared to metal electrodes, which may be translated to higher conductivity of the A5:ETE-PC electrode as compared to metal electrodes (Supplementary Fig. 17b, c).

IRE was applied to U87 cells, and their survival was assessed using live–dead cell staining. Following electroporation, cells were incubated for 4 h to normalize and eliminate any unintended staining caused by reversible electroporation. The results indicated that applying 200 V cm⁻¹ pulses did not result in noticeable cell death, whereas 400 and 600 V cm⁻¹ pulses led to partial reductions in cell viability. Notably, 800 and 1000 V cm⁻¹ pulses caused a complete loss of U87 cell viability (Fig. 2h). SEM imaging was performed to investigate the cancer cell morphology post-IRE. Consistent with the live/dead staining results, cells in the control and 200 V cm⁻¹ groups displayed normal morphology. In contrast, progressive morphological distortion was observed in the 400–1000 V cm⁻¹ treatment groups, with complete cellular destruction evident at 800 V cm⁻¹ and 1000 V cm⁻¹ (Supplementary Fig. 18). A threshold of 800–1500 V cm⁻¹ electric field has been reported to induce IRE in various glioblastoma-derived cells under in vitro conditions[22]. A study on U87 cells demonstrated that applying 90 pulses with 800 V cm⁻¹ strength completely inhibited cell viability[23]. However, the experiment was performed on suspended cells, which are more prone to electroporation than adherent cells due to uniform exposure to the electric field and rounded morphology. On the other hand, our study demonstrates complete ablation of adherent U87 cells by 30 pulses of 800 V cm⁻¹ strength, indicating a higher efficiency. These results indicate that A5:ETE-PC drape extended into the surgical lining of glioblastoma may be able to eliminate residual tumor cells via IRE.

The biocompatibility of ETE-PC was evaluated in human lung fibroblast (HFL1) cells after exposure to increased concentrations of ETE-PC (100, 500, and 1000 μg ml⁻¹) for 24 h. The results from live/dead cell staining showed no reduction in the viability of HFL1 cells, even at the highest tested concentration of ETE-PC (1000 μg ml⁻¹)

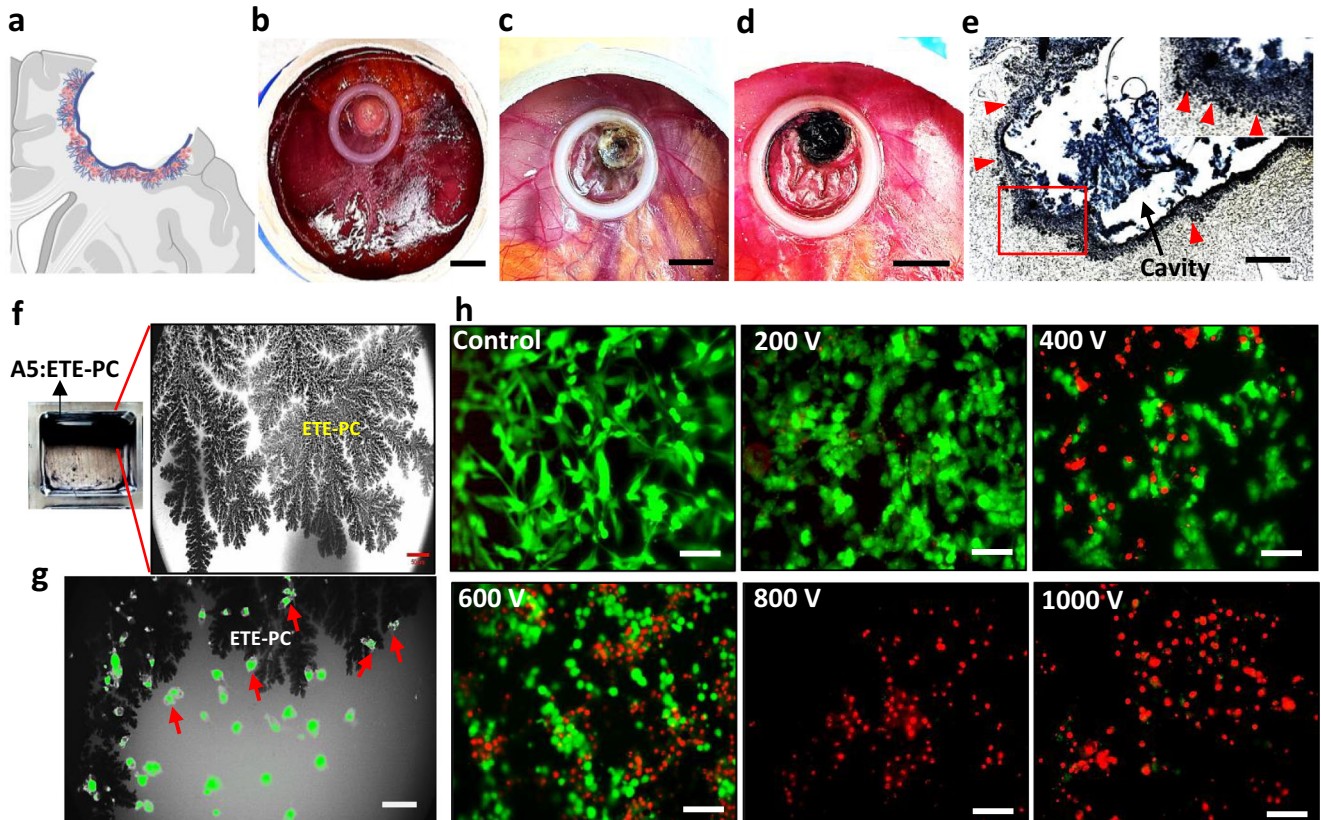

**Fig. 2 | Drape mode for therapeutic use of A5:ETE-PC conductive hydrogel post-surgery. a** Schematic of A5:ETE-PC drape formation in the resection cavity in the brain. Created in BioRender. Yadav, A. (2025) (https://BioRender.com/8vjjsrl). **b**, **c** U87 tumor images before and after surgical resection; scale bar: 5 mm. **d** Image demonstrating resection cavity covered with A5:ETE-PC drape; scale bar: 5 mm. **e** Cryosection (14 μm) of the tumor with resection cavity showing A5:ETE-PC drape lining the cavity wall with ETE-PC dendrites (red arrowheads) extending into the tissue. The enlarged image of the boxed area is in the inset. Scale bar: 200 μm. **f** In vitro simulation of drape model showing A5:ETE-PC layer and ETE-PC dendrites in 0.5% agarose (mimicking brain tissue density) after electrofunctionalization; scale bar: 50 μm. **g** Fluorescence image showing Calcein-AM (live cell dye) stained U87 cells (green) in the vicinity of ETE-PC dendrites. n = 3 independent experiment. **h** Live(green)/dead(red) cell staining of U87 cells subjected to IRE by application of 3 sets (10 each) of 1 ms long high voltage pulses (200–1000 V); scale bar: 100 μm. n = 3 independent experiments.

(Supplementary Fig. 19a). Further, IC50 was calculated by plotting a dose vs. response curve from MTT assay data. The results revealed that ETE-PC has a high IC50 value of around 2650 μg ml$^{-1}$, suggesting its high biocompatibility (Supplementary Fig. 19b). We have previously demonstrated the biocompatibility of A5, showing no significant toxicity after direct exposure to 1 mg ml$^{-1}$ of A5 for 24 h[10]. This finding was further supported by experiments where cancer cells cultured on A5 layers (20 mg ml$^{-1}$) exhibited no signs of toxicity. Instead, the cells grew effectively into a monolayer, indicating that A5 does not inhibit cell proliferation (Supplementary Fig. 20a–c). Further, SEM images of cells grown on the A5 layer demonstrated normal morphology (Supplementary Fig. 20d–k). Overall, these data indicate that both A5 and ETE-PC are highly biocompatible and suitable for tissue injections.

### Intertumoral electrode implantation via endovascular catheter

Injections into deep-seated brain tumors can pose challenges, as they might affect the sensitive, normal brain tissue surrounding them. Here, we demonstrate a catheter-based approach to administering material via blood vessels (Fig. 3a). Endovascular catheters are an established technology routinely used in clinics for endovascular procedures and are amenable to robotic surgeries. In brain tumor embolization, catheters are used to access and occlude the vessels supplying tumors, and catheters capable of breaching the endothelium to treat diseases in the brain parenchyma are well established (trans-vessel wall devices)[24]. We used a nitinol catheter mounted into a Hamilton syringe to insert into a blood vessel in CAM, reaching the tumor (Fig. 3b)[25].

Once the material was injected into the tumor, the catheter was used as an electrode to electrofunctionalize the A5:ETE-PC. The cryosection of the tumor exhibited the catheter track and electrofunctionalized conductive hydrogel (Fig. 3c). Electrical measurements of tumor sections demonstrated a linear voltage-current dependence over varying distances (Fig. 3d). Overall, these results suggest that A5:ETE-PC can be injected and electrofunctionalized into deep-seated tumors via the vascular system using endovascular catheters. This approach will enable localized electrotherapy of glioblastoma without going through the sensitive brain tissue, and potentially reduce the need for open brain surgery.

### A5:ETE-PC electrode-mediated IRE drives immunogenic cell death and immune cell infiltration in the tumor

IRE has been shown to inhibit tumor growth and induce a systemic anti-tumor immune response via immunogenic cell death (ICD)[26,27]. Therefore, we evaluated the potential of A5:ETE-PC electrode-mediated IRE to initiate an antitumor immune cascade. To assess this, we characterized the secretion of damage-associated molecular patterns (DAMPs), including Calreticulin (CRT) and HMGB1, via immunofluorescence in an in vitro model. HMGB1 is a marker of late-stage ICD and plays a crucial role in antigen presentation by activating dendritic cells. During the post-demise phase of ICD, nuclear HMGB1 relocates to the cytoplasm and is subsequently released extracellularly upon plasma membrane rupture. Our results showed distinct cytoplasmic localization of HMGB1 in IRE-treated cells, in contrast to the

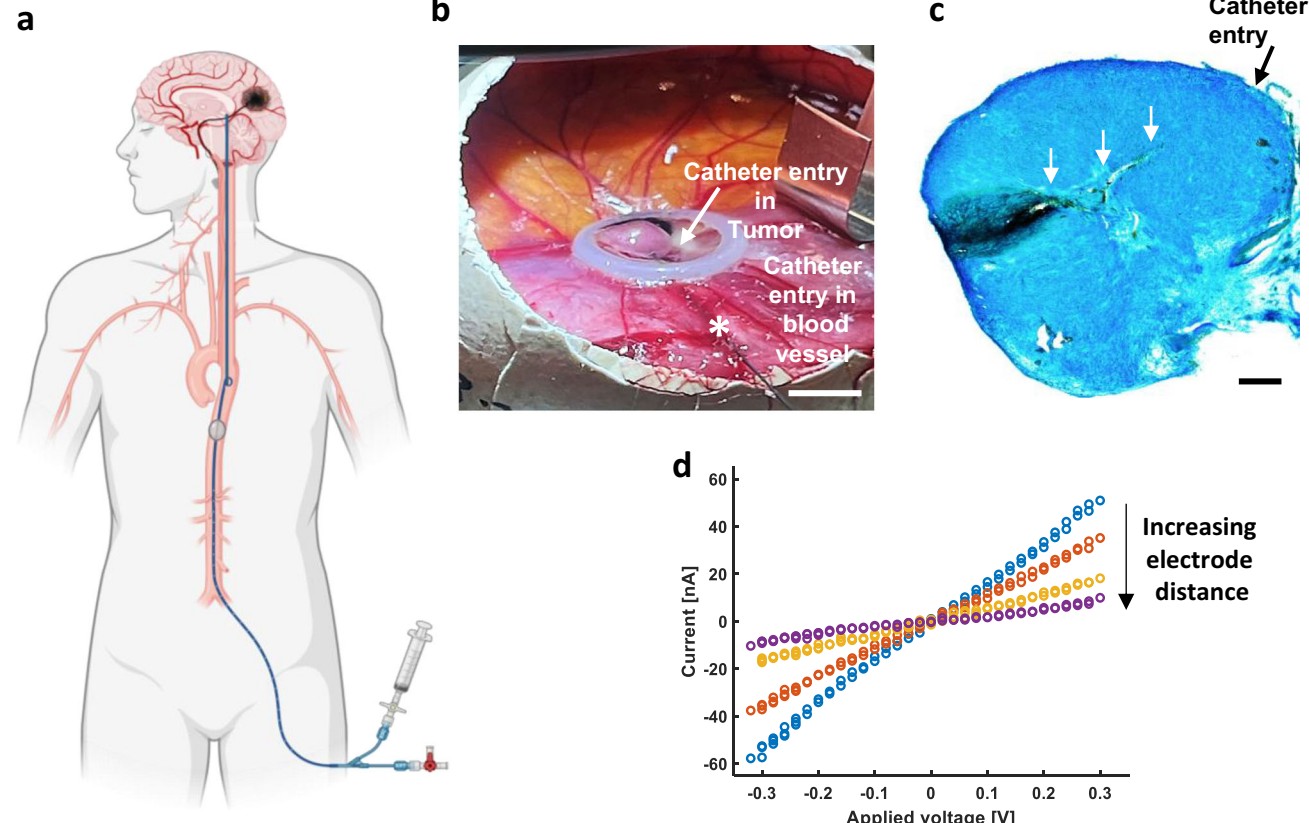

**Fig. 3 | Catheter-based approach of A5:ETE-PC injection in brain tumor.**
**a** Illustration of the catheter-based approach of A5:ETE-PC injection into a brain tumor. Created in BioRender. Yadav, A. (2025) https://BioRender.com/8864ndt.
**b** Photograph showing a catheter reaching the GBM tumor via a blood vessel in CAM and injection of A5:ETE-PC solution in the tumor. *Indicates catheter entry into a blood vessel.; scale bar: 5 mm. **c** Cryosection (30 μm) of injected tumor demonstrating the catheter entry and its track (white arrows) inside the tumor along with A5:ETE-PC hydrogel electrode post electrofunctionalization; scale bar: 500 μm. **d** Current−voltage sweeps were obtained over varying electrode distances in a cryosectioned slice (electrode distance: 15, 80, 140, and 200 μm, respectively). The black arrow defines increasing electrode distance.

control group, where HMGB1 remained confined to the nuclei (Supplementary Fig. 21a). CRT, typically located in the lumen of the endoplasmic reticulum, translocates to the cell membrane following the induction of ICD. Fluorescence imaging revealed a strong membrane-associated expression of CRT in IRE-treated cells compared to control cells. This elevated signal likely reflects CRT accumulation on the cell membrane due to ICD. In contrast, in control cells, CRT remained dispersed within the endoplasmic reticulum (Supplementary Fig. 21b). Hence, these findings indicate that A5:ETE-PC electrode-mediated IRE effectively induces ICD in cancer cells, contributing to its potential as an anti-cancer electroimmunotherapy approach.

To examine the A5:ETE-PC electrode-mediated IRE-induced ICD in vivo, we analyzed the expression patterns of calreticulin and HMGB1 in IRE-treated U87 tumor sections applied with 500−2000 V pulses. Strong calreticulin expression was observed around the A5:ETE-PC electrode in all treatments, likely reflecting calreticulin accumulation on the cell membrane, a hallmark of ICD induction (Fig. 4a). Additionally, HMGB1 showed diffuse red fluorescence in the tissue surrounding the electrode rather than confined to the nucleus, indicating its extracellular release−a characteristic of ICD (Fig. 4b). These effects were consistent across 500−2000 V treatments, with the intensity of the response increasing in dose-dependent manner (Supplementary Figs. 22 and 23). Our results are consistent with previous studies where 80−100 pulses of 1500−3000 V cm⁻¹ induced the release of DAMPs in various cancer models using metal electrodes[26–28]. This suggests that our electrode is on par or even better at inducing IRE and resulting ICD than metal electrodes. To investigate whether electrotherapy using A5:ETE-PC electrode-mediated ICD promotes immune cell infiltration in tumors, we performed CD45 staining on treated tumor sections. CD45 is a pan-leukocyte marker expressed on various immune cells, including dendritic cells, macrophages, T cells, and B cells. The results revealed a high accumulation of CD45+ cells in the tissue near the electrode and around the major blood vessels (arrow heads) indicating significant immune cell infiltration, it also showed an increase going from 1000 to 2000 V (Fig. 4c). It was also correlated with flow cytometry results where a more than two-fold increase in CD45+ cells was observed in IRE-treated tumors compared to control (Fig. 4d).

We further analyzed the immune cell infiltration 72 h post-IRE treatment. Flow cytometric assessment of T lymphocytes and macrophage populations was performed (Supplementary Fig. 24). The results showed a marked increase in both T-cell subtypes−CD4⁺ helper T cells and CD8⁺ cytotoxic T cells in IRE-treated tumors vs control. A significantly higher percentage of CD45⁺CD4⁺ immune cells was observed in IRE-treated tumors (3.34%) compared to controls (0.86%) (Supplementary Fig. 25a). Similarly, CD45⁺CD8⁺ cell population was found to be increased in IRE-treated tumors (3.97%) compared to untreated controls (0.97%) (Supplementary Fig. 25b). We also checked macrophage infiltration using the chicken macrophage marker KUL01, which was also highly elevated in IRE-treated tumors (22.20%) compared to controls (5.10%) (Supplementary Fig. 25c). Further, statistical analysis from multiple independent experiments revealed more than two-fold rise in CD4⁺ and CD8⁺ cell populations in IRE-treated tumors (Fig. 4e, f). Moreover, around a 4-fold increase in macrophages was noticed (Fig. 4g).

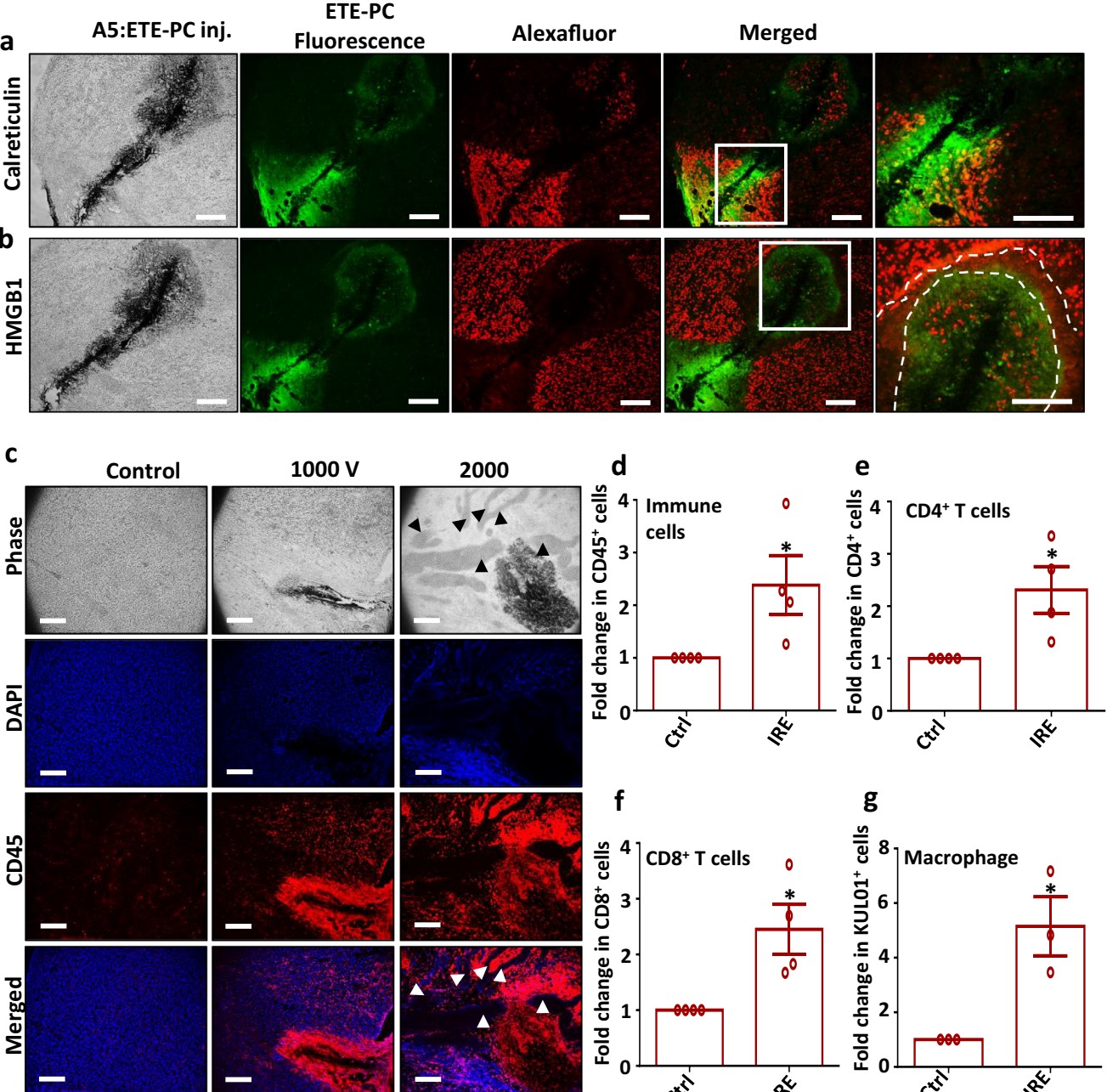

**Fig. 4 | IRE application via A5:ETE-PC electrode induces ICD and immune cell infiltration in tumors.** In situ assembly of A5:ETE-PC electrode was done in in vivo U87 tumors, and IRE treatment was given. Tumors were harvested after 24 h, sectioned, stained, and imaged using a fluorescence microscope. Fluorescence images demonstrate staining of ICD markers **a** calreticulin and **b** HMGB1 in U87 tumor sections (16 μm). ETE-PC monomer fluorescence (green), calreticulin or HMGB1 (red), scale bar: 200 μm. n = 3 independent experiments. **c** Fluorescence images demonstrate staining of the common immune cell marker CD45 in U87 tumor sections (16 μm) treated with IRE. CD45 (red) and nucleus (DAPI, blue). Black and white arrowheads represent major blood vessels. Scale bar: 200 μm. n = 2 independent experiments. **d–g** Immune cell infiltration was examined by flow cytometry as a percentage of CD45+, CD4+, and CD8+ cells along with KUL01+ chicken macrophages in IRE-treated vs control tumors 72 h post-IRE treatment. Bar graphs representing fold increase in CD4+, CD8+, and KUL01+ cells in IRE-treated tumor vs. control. Statistical significance was examined by a one-sample t-test for one-sided analysis of variance among multiple groups. Data indicate mean ± SEM, measurements were taken from distinct samples, n = 4 for CD45 (*p = 0.045), CD4 (*p = 0.030) and CD8 (*p =0.024), n = 3 for KUL01 (*p = 0.031).

These findings suggest that A5:ETE-PC mediated IRE could induce infiltration of immune cells in tumors, indicating elicitation of anti-tumor immune response. A more than two-fold increase in T cells represents substantial recruitment to the tumor microenvironment following treatment. The marked rise in cytotoxic T cells is particularly critical, as these cells directly target and eliminate tumor cells, underscoring the immune-activating potential of electrotherapy. Previous reports also showed a temporal increase in CD8+ T cells in tumor post-IRE in a murine model[4,26,27]. A significant rise in the macrophage population can be correlated with previous studies where a multifold influx of macrophage populations in tumors was observed post-IRE[4,26]. Macrophage influx can be attributed to inflammatory cell death caused by IRE. It could be crucial in the case of glioblastoma, where macrophages account for 30–50% of the tumor mass. Overall, these findings demonstrate that A5:ETE-PC electrode-mediated IRE effectively induces ICD, enhancing immune cell infiltration within the tumor

microenvironment. This supports its potential as an immunomodulatory cancer therapy.

In this study, we report bioresorbable in situ assembled injectable non-substrate-bound conductive hydrogels as highly conductive flexible electrodes for electroimmunotherapy of glioblastoma. Despite being used in clinics for pancreatic and prostate cancer, electrotherapy is not included in new approaches to the localized therapy of glioblastoma. Due to the high invasiveness and surgical dependence of current techniques, IRE has not reached its true potential in glioblastoma therapy. Our soft electrodes (injectable and drape electrodes) match brain softness and avoid the implantation and removal of the electrodes using surgery. The flexible and branched nature of the drape electrode enhances the tissue electrode interface significantly, leading to better conductive properties and higher therapeutic coverage in tumors. Further, these electrodes can be implanted in various ways, providing a flexible and multimodal therapeutic approach. Moreover, using these electrodes for IRE therapy impairs tumor growth and elicits an anti-tumor immune response via ICD (Supplementary Fig. 26). Most importantly, when combining CED with IRE, these electrodes can ablate the tumor completely. The endovascular implantation and treatment by CED-IRE combined with the drape technology, the latter for a subsequent targeting of invasive cells and recurrent glioblastoma at the tumor margin, offers a novel potential strategy for treatment and potentially reduces the need for open brain surgery. This electrode technology represents a promising and versatile electrotherapy platform for treating glioblastoma and other solid tumors in sensitive tissue environments.

## Methods
### Animal ethics
Every experiment involving animals have been carried out following a protocol approved by an ethical commission. This study was conducted in accordance with the national legislation of Sweden and with European Community guidelines for animal studies. All procedures were approved by the ethical committee in Malmö–Lund (5.8.18-19103/2023).

### Maintenance of cell lines
Human glioblastoma (U87) and human normal lung fibroblast (HFL1) cell lines were maintained in DMEM (Thermo Fisher, Gibco, Cat no. 11995073) supplemented with 10% FBS (Thermo Fisher, Gibco, Cat no. 16000044), 100 units penicillin and 100 $\mu$g ml$^{-1}$ streptomycin (Thermo Fisher, Gibco, Cat no. 16000044) and 1% NEAA (Thermo Fisher, Gibco, Cat no. 11140035) in a humidified $CO_2$ incubator at 5% $CO_2$ and 37 °C. U87 cells were kindly shared by Dr. Johan Bengzon (Lund University), while HFL1 cells were provided by Dr. Sara Rolandsson (Lund University).

### Evaluation of A5:ETE-PC 3D flexible electrode in agarose
A5 and ETE-PC were synthesized as previously reported. The A5:ETE-PC electrode was assembled in agarose by injecting the A5:ETE-PC nanoparticle solution into agarose, followed by electrofunctionalization, as described in the reported procedure. Briefly, a solution of A5 (20 mg ml$^{-1}$) and ETE-PC (40 mg ml$^{-1}$) in Milli-Q water was injected in 0.5% agarose (Agarose, low gelling temperature, Sigma Aldrich, Cat no. A9414-25G) mold in PBS (pH 7.4) and ETE-PC was allowed to diffuse. After 15 min an Au-coated W-electrode was connected to the A5:ETE-PC self-assembled structure functioning as the working electrode (anode), while the counter electrode (cathode) was put in the agarose. ETE-PC was electrofunctionalized on A5 using 2 V vs. Au-coated counter electrode (Keithley sourcemeter 2612B, Keithley Instruments). The electrode formation in agarose was examined using bright-field microscopy (Nikon, 4× objective).

The functionalization of ETE-PC on A5 and electrochemical properties of A5 alone vs. A5:ETE-PC electrode were analyzed by cyclic voltammetry (CV) (0–1 V range, scan rate–0.01–0.05 V s$^{-1}$) and electrochemical impedance spectroscopy (EIS) (for the frequency range of 1 Hz–100 kHz) using Autolab PGSTAT204 potentiostat (Metrohm) by using a three-electrode system where the A5:ETE-PC injection was connected to an Au-coated metal electrode serving as the working electrode, while another Au-coated metal electrode embedded in the agarose gel acted as the counter electrode. Ag/AgCl was used as the reference electrode.

### CAM tumor model
Fertilized chicken (*Gallus gallus*) eggs were placed in a specialized egg incubator at 37.5 °C with 65% relative humidity. The day the eggs were placed in the incubator was designated embryonic development day 0 (EDD0). On EDD4, a 2 cm wide window was opened into the shell where the air sac was located using a mechanical egg opener to expose the CAM. The window was covered with a 30 mm petri dish, and eggs were put back into the incubator. To develop an in vivo tumor model, $1.5 \times 10^6$ U87 human glioblastoma cells in DMEM were mixed with Matrigel (Fisher, Corning, Cat no. 354234) in a 1:1 ratio (total volume 40 $\mu$l) and implanted on the CAM at a site where big blood vessels with significant branching were present on EDD9. The tumor growth was observed, and images of tumors were taken. All experiments were conducted within EDD0 to EDD20, and in all experiments except flow cytometry the chick embryos were euthanized on EDD18.

### Implantation of A5:ETE-PC hydrogel electrode in in vivo U87 tumor and application of IRE
U87 tumors generated on the CAM were injected with a mixture of A5:ETE-PC (20:40 mg ml$^{-1}$) using a metal (Pt-Pd)-coated glass capillary (30 $\mu$m diameter), and an initial electrofunctionalization was performed by applying a 1.2 V bias for 2.5 min using the coated capillary as working electrode against a Pt-Pd counter electrode touching the tumor surface. ETE-PC was then allowed to diffuse in tumor tissue for 5 min, followed by a second electrofunctionalization by applying 3 V for 10 min. After that, tumors were harvested and fixed in 4% paraformaldehyde for 24 h at 4 °C and washed with PBS, followed by incubation in 30% sucrose solution for cryoprotection. The tissues were frozen on dry ice in Tissue-Tek OCT (Fisher Scientific: Epredia Neg-50). Frozen tumors were cryosectioned (16 or 30 $\mu$m section thickness, dependent on subsequent processing) using Cryostar NX70 cryostat and mounted on Superfrost Gold microscope slides for microscopy or interdigitated gold electrodes for conductivity measurements. Images were taken using a bright field microscope. During the experiment, EIS was performed on the tumor to compare the impedance with metal vs in situ assembled A5:ETE-PC flexible electrodes. The conductivity of the formed hydrogel electrode was measured in tumor sections (30 $\mu$m) placed on interdigitated gold electrodes by sweeping voltage from −0.2 to +0.2 V and registering current. The in situ assembled A5:ETE-PC electrodes connected to metal coated capillary were used to induce IRE by applying 3 sets of 300 $\mu$s long pulses (200 pulses each set) of 150 V (effective electric field strength 1000–1500 V cm$^{-1}$).

### Convection-enhanced delivery of A5:ETE-PC
To perform convection-enhanced delivery of A5:ETE-PC in U87 tumor, a Hamilton syringe with a thin needle (ga33, 108 $\mu$m inner diameter) was filled with A5:ETE-PC solution (20:40 mg ml$^{-1}$ in milliQ water) and connected a syringe pump (KD Scientific, USA). The solution (5 $\mu$l) was injected into the tumor at an injection rate of 10 $\mu$l/h. Post injection, the material was electrofunctionalized by applying a bias of 1.5 V for 10 min using a syringe needle as the working electrode (anode) and Ag/AgCl as a counter electrode (cathode). Tumors were harvested, fixed in 4% PFA, and processed for cryosectioning.

## Tumor growth studies after IRE treatment

To examine the effect of polymer-mediated IRE on tumor growth inhibition, A5:ETE-PC hydrogel electrodes (electrode spacing 1–1.5 mm) were implanted either via capillary injection or convection-enhanced delivery in U87 tumors on CAM. These electrodes were used to apply IRE [3 × 200, 150 V (1000–1500 V cm$^{-1}$), 500 μs long pulses] on EDD15. Following IRE, tumor dimensions were measured daily using a vernier caliper until EDD18. Tumor volumes were calculated from these measurements using the following formula;

Tumor volume (V) = $0.5 \times (L \times W^2)$, where L = length and W = width of the tumor.

On EDD18, tumors were excised, imaged, and weighed. Tumor volume and weight were analyzed statistically and represented as graphs. Statistical significance was analyzed by performing an unpaired t-test in Graphpad Prism (Version 6.01).

## Surgical resection and drape formation

To mimic the post-surgery tumor resection cavity in the brain, GBM tumors on CAM were surgically removed using electrosurgical equipment (Diatermo MB 160, GIMA), leaving a resection cavity. For drape formation, the cavity was filled with A5:ETE-PC solution (20:40 mg ml$^{-1}$ in milliQ) and incubated for 15 min. The excess solution was removed, and electrofunctionalization was performed by applying 1.2 V bias (Keithley sourcemeter 2612B, Keithley Instruments) using a Au-coated W-needle electrode as the working electrode (anode) and Ag/AgCl as counter electrode (cathode) for 15 min. EIS measurements were taken before and after drape formation using a two-electrode setup with an Au-coated needle electrode as the working electrode and Ag/AgCl as the counter electrode. Additional distance-dependent electrical measurements on the drape were performed between two Au-coated needle electrodes placed 1–4 mm apart. An applied voltage was swept using a Keithley sourcemeter 2612B (Keithley Instruments), and the resulting current was registered. Post-drape formation, tissue with a resection cavity was harvested and processed for cryosectioning as described earlier. Sections (14 μm) were mounted on Superfrost Gold microscope slides for microscopy. Sections were imaged with an Olympus BX53 microscope with 4×, 10×, and 20× objectives.

## In vitro IRE of cancer cells via A5:ETE-PC hydrogel

The efficacy of A5:ETE-PC flexible electrode in electrotherapy of cancer was examined by performing IRE in a cell-based in vitro model of glioblastoma in an agarose mold. Briefly, U87 cells (5 × 10$^4$ cells/well) were grown in 8 well chamber slide (μ-Slide 8 well$^{high}$ ibiTreat, ibidi). After 24 h, 200 μl of 0.5% agarose (low gelling temperature in DMEM) was added to wells. Once the agarose gelled, a solution of 20 μl A5:ETE-PC (A5 (20 mg ml$^{-1}$) and ETE-PC (40 mg ml$^{-1}$) in Milli-Q water) was added to one side of the well, and ETE-PC was left to diffuse laterally into the agarose. After 15 min, A5 was electrofunctionalized with ETE-PC to form flexible electrodes by applying a bias of 2 V vs Au-coated counter electrode (Keithley sourcemeter 2612B, Keithley Instruments). The electrode formation was examined by bright field microscopy (Olympus, 10× objective). To perform IRE, high voltage pulsed electric fields were applied (MicroPulser, Bio-Rad) to cells using the A5:ETE-PC electrode. Specifically, cells were treated with a series of pulsed electric fields (200–1000 V cm$^{-1}$). A sample without A5:ETE-PC electrode and any treatment was used as a control. After treatment, cells were allowed to normalize for 4 h in an incubator at 37 °C. Later, live/dead cell staining was performed by incubating cells with calcein-AM (Thermo Fisher, Invitrogen, Cat no. C3100MP, 2 μM in PR-free DMEM) and PI (Thermo Fisher, Invitrogen, Cat no. P1304MP, 2 μM in PR-free DMEM) dyes for 30 min, followed by examination of live/dead cells using fluorescence microscopy. The morphological changes in cells post-IRE were examined by scanning electron microscopy (SEM). For SEM, cells were fixed in 4% PFA followed by rinsing in PBS. An ethanol exchange series of 10 min each in 30, 50, 75, 90, 95, and 99.5% ethanol

was used. Samples were left to air-dry before being coated with 4 nm Pt/Pd in a Quorum K850 CPD sputterer (Quorum Technologies) and imaged using a cold-field emission SEM (SU8010, Hitachi) operating at 10 kV.

## In vitro toxicity and IC50 of ETE-PC in normal lung fibroblast (HFL1)

HFL1 cells were seeded in 96-well plates at a density of $2 \times 10^4$ cell per well and grown for 24 h. Later, cells were treated with various concentrations of ETE-PC (100–1000 μg/ml) for 24 h. Post-treatment, 200 μl calcein-AM (2 μM in PR-free DMEM) and PI (2 μM in PR-free DMEM) dyes were added to wells and incubated for 30 min, followed by an examination of live/dead cells using fluorescence cell imaging (Spark Cyto cell imager, Tecan). The half-maximal inhibitory concentration (IC50) of ETE-PC in HFL1 cells was calculated by MTT cytotoxicity assay. HFL1 cells grown in 96-well plates were treated with increasing concentrations of ETE-PC (0.1–10 mg/ml) for 24 h. Post-treatment, cells were incubated with 200 μl of MTT solution (0.5 mg/ml in PBS) for 4 h. After incubation, MTT was removed, and the resulting formazan crystals were dissolved in 200 μl of propanol. Absorbance was recorded at 570 nm using Spark Cyto plate reader (Tecan). A dose (log concentration) vs response curve was plotted using acquired data to determine the IC50 via GraphPad Prism software (Version 6.1).

## Intertumoral catheter injection via a blood vessel

To generate the injection catheters, nitinol tube with outer diameter 0.203 mm (Euroflex, NiTi (287) SE 508) was cut in 15 cm sections between two alumina plates (top 0.5 mm, bottom 2 mm) using electrical discharge machining (EDM, Brother HS50-A, wire Ø0.25 mm, 60 V, 20 us discharge time, approximately speed 6 mm/min) generating one 90-degree blunt end and one 45-degree pointy end. The blunt end was mounted into the hub (metal and rubber) of a 32 G Hamilton RN needle (replacing the existing needle) and glued in place with epoxy glue. The catheter was then mounted into a 10 mL Hamilton syringe (701 RN).

The chicken CAM model with an implanted U87 tumor was used for the experiments. The syringe with the catheter was mounted into a micromanipulator, and the syringe was loaded with A5:ETE-PC (20:40 mg ml$^{-1}$) (6 μL). The catheter was inserted through a blood vessel in the CAM toward the tumor. The epithelium was penetrated close to the tumor, and the catheter was inserted into the tumor tissue. The polymer mixture was then injected by hand for one minute. The catheter was connected to the anode, and the cathode was connected to the CAM membrane using a Nickel blade electrode (100 × 8 × 0.15 mm), ensuring a large contact area. After 5 min, the material was electrofunctionalized (1.5 V bias) for 5 min using a Keithley SourceMeter 2612B (Keithley Instruments). The tumor was isolated from the CAM membrane, washed in PBS 7.2, and fixated in PFA (4%). Next, the tumor was rinsed with phosphate-buffered saline (PBS) pH 7.4, cryoprotected in PBS containing 25% (w/v) sucrose (Sigma, >99.5% BioXtra S7903) and 20% (v/v) NEG 50 $^{TM}$ OCT (Epredia, 6502), frozen in NEG 50 $^{TM}$ OCT on dry ice and cryosectioned. Serial 30-μm-thick cryostat sections were stained with 0.05% Methylene Blue aqueous solution (Sigma-Aldrich, 319112) for 25 min, differentiated with 96 and 70% EtOH, and rinsed in PBS. All stained slides were coverslipped using glycerol-PBS (8 + 2) as a mounting medium. The stained sections were examined with an Olympus BX53 microscope using UPlanSApo objectives (4×/0,16; 10×/0,40; 20×/0,75) and CellSens Dimension software. Overview images with the 4× objective were obtained using the manual Instant Multiple Image Acquisition function. Raw image files (vsi format) were transformed into TIFF files using Fiji/ImageJ, and the color balance was adjusted in Adobe Photoshop 2025.

Some 35-μm-thick cryostat sections were collected on interdigitated Au electrodes connected to a Keithley sourcemeter 2612B

(Keithley Instruments). Two of the interdigitated electrodes were contacted using external microelectrodes. An applied voltage was swept, and the resulting current was registered. This was repeated for differently spaced interdigitated electrode leads at 80–200 µm separation. The distance between adjacent electrodes was 15 µm, and the width was 2.5 mm. The voltage was swept from −0.3 V to +0.3 V at 10 mV s$^{-1}$.

## Immunofluorescence studies

To examine the effect of IRE on induction of Immunogenic cell death (ICD) in U87 cells in vitro, the expression of ICD markers calreticulin and High mobility group box 1 (HMGB1) were analyzed by immunofluorescence. U87 cells were given IRE treatment using A5:ETE-PC electrode. Post-treatment, cells were fixed in 4% paraformaldehyde (PFA) for 30 min at 37 °C. After fixation, the agarose layer was removed by melting in a preheated water bath to recover cells. The cells were permeabilized with 0.1% Triton X-100 for 10 min at room temperature (RT), followed by PBS wash. The cells were then blocked with 10% normal goat serum for 1 h at RT and incubated with calreticulin (1:500 dilution, Rabbit anti-human, ab92516, Abcam) and HMGB1 (1:500 dilution, Rabbit anti-human, ab18256, Abcam) primary antibodies overnight at 4 °C. Subsequently, cells were rinsed with PBS and incubated with goat anti-rabbit IgG-Alexa Fluor 546 (Invitrogen A11035) diluted 1:500 for CRT and HMGB1, respectively, for 1 h at RT. After buffer rinses, the cells were mounted with ProLong Antifade Gold with DAPI (Thermo Fischer P36931) as the mounting medium.

Immunofluorescence studies on U87 tumor sections that had undergone IRE were performed to examine the expression of Ki67, HMGB1, calreticulin, and CD45. Briefly, U87 xenografts were developed on CAM and given IRE treatment as described above. Twenty-four hours post-treatment, tumors were processed for cryosectioning. Tumor sections (16 µm) were incubated with Ki67 (1:800 dilution, Rabbit anti-human, ab15580, Abcam), calreticulin (1:500 dilution, Rabbit anti-human, ab92516, Abcam) and HMGB1 (1:500 dilution, Rabbit anti-human, ab18256, Abcam) and CD45 (1:200 dilution, mouse anti-chicken, 7500970, Thermo Fisher) primary antibodies overnight at 4 °C. Subsequently, cells were rinsed with PBS and incubated with goat anti-rabbit IgG-Alexa Fluor 546 (Invitrogen A11035) for Ki67, calreticulin, HMGB1, and goat anti-mouse IgG-Alexa Fluor 546 (Invitrogen A11030) for CD45, respectively, (diluted to 1:500) for 1 h at RT. After buffer rinses, the sections were mounted with ProLong Antifade Gold with DAPI (Thermo Fisher P36931) as the mounting medium.

## Flow cytometry analysis

Following IRE, flow cytometric analyses were performed to study the immune cell infiltration into the tumor. Briefly, tumors were excised at 72 h post-IRE and manually chopped in DMEM using a scalpel. The chopped pieces of tumors were then manually dissociated by pressing them against a 40 µm strainer (pluriStrainer Mini 40 µm, pluriSelect) fixed on a 1.5 ml microcentrifuge tube using a syringe plunger to pass it through. The dissociated cells were then collected by centrifugation at 600 $g$ for 5 min at 4 °C. Cells were resuspended in FACS buffer (10% FBS, 2.5 mM EDTA, 0.05% Sodium azide in PBS) and incubated with fluorophore conjugated primary antibodies (1:100 in FACS buffer) for CD45 (APC conjugated mouse anti-chicken, MA5-28677, Thermo Fisher), CD4 (PE conjugated mouse anti-chicken, MA5-28686, Thermo Fisher), CD8 (PE conjugated mouse anti-chicken, MA5-28726, Thermo Fisher) and KUL01 (PE conjugated mouse anti-chicken, MA5-28828, Thermo Fisher) respectively for 30 min on ice. After 30 min, cells were washed with FACS buffer twice by centrifugation at 600 × $g$ for 5 min at 4 °C. Cells were then fixed in PFA (4% in PBS) for 10 min on ice. Post-fixation, PFA was removed, cells were resuspended in FACS buffer, and stored at 4 °C till analysis. Data were acquired using BD Symphony A1 flow cytometer (BD Biosciences) and analyzed by FlowJo software (v10.10.0, FlowJo LLC, BD Biosciences).

## Statistical analysis

Statistical analyses for all experiments were conducted using Graph-Pad Prism (v6.01). No statistical method was used for a predetermined sample size. In vitro experiments were not performed in a blinded fashion, but were measured with objective methodologies. The statistical tests used to assess significance, along with biological and experimental replicates in each dataset, are specified in the figure legends for each corresponding figure. Statistical significance was defined as $P \leq 0.05$.

## Reporting summary

Further information on research design is available in the Nature Portfolio Reporting Summary linked to this article.

## Data availability

All data supporting the findings of this study are available within the article and its supplementary files. Any additional requests for information can be directed to and will be fulfilled by the corresponding authors. Source data are provided with this paper.

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

## Acknowledgements

This study was conducted within the Lund University Strategic Research Areas, MultiPark and NanoLund. We thank Dr. Xenofon Strakosas for assistance with interdigitated electrodes and Damien Hughes for assistance with coating the injection capillaries. Equipment within Lund Nano Lab (LNL) was used to enable this research. R.O. acknowledges funding from the Swedish Cancer Society (24 3629 Pj) and the Swedish Research Council (2023-04965). M.H. acknowledges the Swedish Science Council should grant no. 2021-05231. J.B. acknowledges funding from the Swedish Cancer Society grant CAN 23 2937 Pj 01 H, the Swedish Childhood Cancer Foundation grant PR2022-0117, the Fru Berta Kamprad Foundation, and the Sjöberg Foundation.

## Author contributions

Conceptualization: R.O. and A.S.Y. Methodology: by A.S.Y., U.A., M.H., P.E., F.E., and R.O. Design and synthesis: R.O., A.H.M., and M.A.S. Chicken experiments: A.S.Y., U.A., K.H., P.E., and F.E. Cell work: A.S.Y., K.H., U.A., and J.L. Electrical characterization: A.S.Y. Mechanical characterization: C.D., A.S.Y., and U.A. Funding acquisition: R.O., J.B., and M.H. Supervision: R.O. Manuscript writing: A.S.Y. and R.O.

## Funding

## Competing interests

R.O. has filed a patent application covering the chemical compositions. The remaining authors declare no competing interests.
