## [Transparent Peer Review File · Nature Communications]

Injectable Bioresorbable Conductive Hydrogels for Multimodal Brain Tumor Electroimmunotherapy

Corresponding Author: Professor Roger Olsson

Version 0:

Reviewer comments:

Reviewer #1

(Remarks to the Author)

This study presents an injectable conductive hydrogel that can be polymerized in situ within glioblastoma (GBM), serving as an electrode for irreversible electroporation therapy. The hydrogel features a unique structure, with ETE-PC dendrites extending from an A5 core – both intrinsically conductive materials- resulting in a high conductivity and low electrode impedance. The dendritic morphology further enhances performance by increasing surface area and facilitating deep tumor penetration. The hydrogel is applied using three different delivery methods (injection, catheter, drape) and evaluated through in vitro and in vivo experiments in chicken tumor model.

Comments and suggestions

1. Electrofunctionalization of ETE-PC: The authors claim that the ETE-PC effectively forms a dendrite structure protruding from A5 core. However, according to reference 11, ETE-PC fails to generate a stable structure with A5, as it tends to disperse the A5 core structure prior electrofunctionalization. The manuscript must address this discrepancy and clarify how the instability issue reported in prior literature is mitigated in the present study. (Page 6, line 172) Additionally, while the authors state that the concentration of ETE-PC influences the oxidation potential, there is insufficient data to demonstrate how varying concentrations impact the formation and morphology of the dendritic structure. More detailed experimental evidence is necessary to support this claim.
2. Validation of the hydrogel property of A5:ETE-PC: The authors deliver the idea of hydrogel both to the in-situ electrofunctionalized core and dendrite through electrofunctionalization, as the chemical precursors were delivered as a water-soluble mixture. However, for precision on the terminology, we recommend the authors revisit the hydrogel properties of A5:ETE-PC. In general, hydrogels consist of a large amount of water content within its matrix innately, and the water content could hinder or enhance the characteristics or the performance of the material. We recommend the authors investigate the water content of both A5:ETE-PC core and dendrite, while comparing the impact of water content on its mechanical and electrical properties, as the material is naturally surrounded by a biofluid. Moreover, as most of injectable hydrogels do, the zwitterionic nature of the polymer network of A5:ETE-PC could suffer from excessive swelling in longer-term applications, while also under its gradual biodegradation; thus, unless the electrophoretic therapy could completely eliminate the tumor immediately only with a single-term usage, it is inevitable that the performance should be prone to two significant properties that would affect simultaneously.
3. Bioresorbable property of A5:ETE-PC hydrogel: The claim that the ETE-PC hydrogel is bioresorbable lacks adequate experimental data. Both A5 and ETE-PC contain thiopene rings that are generally considered not to be hydrolysable, raising doubts about their resorbability. Although reference 10 claims that A5 may exhibit superior bioresorption than PEDOT: PSS, this comparison is inadequate, as PEDOT:PSS is widely recognized as non-bioresorbable. Referencing prior work that does not directly address the degradation kinetics of A5, or ETE-PC is insufficient to support the bioresorbable nature of the material. To claim the biocompatibility of the bioresorption of A5:ETE-PC, we recommend the authors to further investigate the biocompatibility of the degradation byproduct, especially regarding the ability of its compatibility with the natural metabolic mechanism.
4. Biocompatibility of A5:ETE-PC hydrogel: The authors state that both A5 and ETE-PC are biocompatible (page 13, line 380-382). However, if the hydrogel is indeed bioresorbable as claimed, its biocompatibility must also account for the concentration and nature of its degradation byproducts. Toxicity is often dose-dependent and can arise from accumulated degradation products, even if the parent materials are considered biocompatible. Therefore, additional experimental data are needed to assess how varying concentrations of A5 affect cell viability or tissue response.
5. Irreversible electroporation (IRE) efficacy: The authors claim that IRE using the hydrogel achieves effective tumor ablation,

enhanced by immunogenic cell death and increased cell infiltration. However, the manuscript lacks direct evidence demonstrating the effect of IRE on normal cells. To validate the therapeutic efficacy of the IRE treatment, an additional control group – consisting of normal cells treated with and without IRE-should be included.

6. Homogeneity of dendrite structure: The authors claim that the ETE-PC dendrite structure shows ‘uniform conductivity’ throughout the entire network, as all chemical compositions (A5, ETE-PC) are intrinsically conductive (page 9, line 261). However, this simple inference does not validate reality, as there still exists possibility of local structure formations that could hinder the uniformity of conductivity, such as local electrostatic aggregation, microphase separation, etc. Regarding the issue, we recommend the authors provide detailed information on the chemical and physical properties of the in-situ formed dendrite microstructures, compared to the ‘core’ of A5:ETE-PC hydrogel, especially regarding the chemical homogeneity, mechanical modulus and adhesiveness. Although dendrite itself holds the core significance of the following findings, there exists limited evidence regarding the issue.

7. Control experiments on electrical characterization: The authors repeatedly claim that A5:ETE-PC shows superior electrical properties by its enhanced interfacing (increase in surface area due to dendrite formation), referring to its decrease in impedance compared to mere A5 or metal electrodes (lines 192 (page 6), 215 (page 7), figures S3(e), and S7(b)).

However, to highlight the sole effect of the dendrite structure on impedance, the authors should compare the provided data with a bulk A5:ETE-PC hydrogel. Assuming there does exist strong and stable interface between A5:ETE-PC dendrite or bulk and tumor area, a controlled ex-vivo impedance analysis would be a suggestable setup for further investigation.

8. Adhesion and mechanical stability of A5:ETE-PC: The authors claim that the dendrite structure forms enrichment in interfacing, resulting in lower impedance and enhanced electrophoresis efficacy. However, no evidence exists on the mechanical stability of the dendrite – tumor tissue interface, which in turn could critically impact the actual IRE therapy. Provided in the prior studies with similar chemical compositions (reference 10, 11), the shear moduli of are extremely low, initially to enable the mechanical match with benign brain tissue. However, such low moduli could further act as the bottleneck on stable interfacing in vivo. Authors should provide detailed information about the stability of interfacing of dendrites and tumor cells, using indentation methods.

9. Time-resolved analysis of electrofunctionalization: The authors only visualizes the already-formed A5:ETE-PC after the electrofunctionalization (page 7, lines 212-214, figure S5). To further ensure that the polymerization step takes a short amount of time that minimally affects the physiological activity of the surrounding benign tissues, we recommend the authors provide the time-resolved visualization of electrofunctionalization in situ. Additionally, the provided visualization (figure S5, S6) does not precisely mimic the actual environment of electrofunctionalization, as here the process occurs on the scaffold of A5, rather than the injected solution of precursors and its core. Moreover, authors should provide detailed information on the electrofunctionalization process, as there exists no clear elaboration on why the process is divided into two steps (step 1: 1.2V for 2.5 minutes, step 2: 3V for 10 minutes).

10. (Page 9, lines 267–269) Please explain the terminology EDD, as it initially uses the term here.

11. (Page 13, line 360) Please reassign the figure that matches the explanation “Notably, 800 V cm⁻¹ and 1000 V cm⁻¹ pulses 360 caused a complete loss of U87 cell viability”, rather than figure 1h.

Reviewer #2

(Remarks to the Author)

In this paper, the authors developed a new electroimmunotherapy for glioblastoma using an injectable conductive hydrogel. Combination of the injectable conductive hydrogel and the irreversible electroporation (IRE) method significantly reduces the invasiveness of the patients compared to conventional treatments. Moreover, when combined with surgery, it is also effective in eradicating remaining tumor cells, suggesting the potential for becoming a groundbreaking treatment. The same material (A5: ETE-PC) as in the previous study (reference 10) was used, and although there is little novelty in the material, the validity and effectiveness of the evaluation using a chicken chorioallantoic membrane (CAM) tumor model (glioma) are demonstrated. However, several parts are difficult for readers outside the field to understand, and I am convinced that the value of this paper will be further increased by revising it with the following points in mind.

1) There is a lack of information on how the material used (A5: ETE-PC) was optimized. It would be desirable to supplement the process by which the structure, molecular weight, concentration, and A5/ETE-PC mixing ratio were determined to be used.

2) Considering the molecular structure, it is unlikely that the polymer used will be degraded in the body. I presume that the term "bioabsorbable" refers to the disappearance of this polymer from the site of injection into the body. However, information on the pharmacokinetics of these polymers after injection is essential. Information on the organ distribution and excretion from the body after administration into the mammalian brain is required.

3) A voltage was applied, but a clear diagram of how this works is necessary. A clear diagram of the agarose system was attached; however, it would be even more beneficial if there were a diagram that clearly illustrates the relationship between the electrodes and the gel injection site in the CAM system.

Version 1:

Reviewer comments:

Reviewer #1

(Remarks to the Author)

I have some additional suggestions for perfecting this study with robust supporting data.

Additional suggestions:

- Although electrofunctionalization occurs within the area of tumor tissue, it is still vaguely described that only the tumor tissue is specifically targeted as the functionalization region and limited with its propagation or actual function on surrounding normal tissues, in a spatial-wise point of view. Although extremely difficult to formulate an in-situ imaging method for such, it would be much more impactful and intuitively persuasive in terms of the biocompatibility of the novel process. One simple suggestion is to perform histological analyses on both types of tissues after the treatment and show the compatibility of the treatment on the normal region.
- Although repeated claims about the existence of reports on the Young's modulus of A5:ETE-PC as 1-4 kPa, there consistently seems no specific data shown in this article and the referenced paper (Hjort, M., Mousa, A.H., Bliman, D. et al. In situ assembly of bioresorbable organic bioelectronics in the brain. Nat Commun 14, 4453 (2023)). I strongly recommend the authors show the direct measurements of Young's modulus, with results from indentations. It would much more easily support the claim with the benefit of low mechanical stiffness to adhesive interfacing with tumor tissues, when the actual Young's modulus data and relationship with adhesion is displayed in parallel, with appropriate control group.

Reviewer #2

(Remarks to the Author)

In the revised paper, the authors developed a new electroimmunotherapy for glioblastoma using an injectable conductive hydrogel. Combination of the injectable conductive hydrogel and the irreversible electroporation (IRE) method significantly reduces the invasiveness of the patients compared to conventional treatments. Moreover, when combined with surgery, it is also effective in eradicating remaining tumor cells, suggesting the potential for becoming a groundbreaking treatment. The same material (A5: ETE-PC) as in the previous study (reference 10) was used, and although there is little novelty in the material, the validity and effectiveness of the evaluation using a chicken chorioallantoic membrane (CAM) tumor model (glioma) are demonstrated.

The authors have responded sincerely and fully to all reviewer comments, and the content and value of this paper have been significantly improved. Therefore, I have determined that the paper can be accepted as is.

Version 2:

Reviewer comments:

Reviewer #1

(Remarks to the Author)

I think now the manuscript is well prepared for the publication with robust explanation for reviewers' comments.

REVIEWER COMMENTS

We thank the reviewers for carefully reading through and helping us to improve the manuscript.

Reviewer #1 (Remarks to the Author):

This study presents an injectable conductive hydrogel that can be polymerized in situ within glioblastoma (GBM), serving as an electrode for irreversible electroporation therapy. The hydrogel features a unique structure, with ETE-PC dendrites extending from an A5 core – both intrinsically conductive materials- resulting in a high conductivity and low electrode impedance. The dendritic morphology further enhances performance by increasing surface area and facilitating deep tumor penetration. The hydrogel is applied using three different delivery methods (injection, catheter, drape) and evaluated through in vitro and in vivo experiments in chicken tumor model.

Comments and suggestions

1. Electrofunctionalization of ETE-PC: The authors claim that the ETE-PC effectively forms a dendrite structure protruding from A5 core. However, according to reference 11, ETE-PC fails to generate a stable structure with A5, as it tends to disperse the A5 core structure prior electrofunctionalization. The manuscript must address this discrepancy and clarify how the instability issue reported in prior literature is mitigated in the present study. (Page 6, line 172) Additionally, while the authors state that the concentration of ETE-PC influences the oxidation potential, there is insufficient data to demonstrate how varying concentrations impact the formation and morphology of the dendritic structure. More detailed experimental evidence is necessary to support this claim.

Response: We acknowledge and appreciate that the reviewer has taken the time to read our previous publications. In addition to this work, we have previously demonstrated that A5:ETE-PC also forms a conductive structure with dendrites in vivo in brain tissue, and in the tailfin of a zebrafish (<https://doi.org/10.1038/s41467-023-40175-3>). From a hydrodynamic point of view, both the brain and tumors are relatively passive tissues, while the beating heart is more dynamic, which gives a shorter time frame for electrofunctionalization. We hypothesized in ref 11 that, after initial stabilization of the core by self-aggregation, the time frame for further stabilization was too short in the vicinity of the beating heart. ETE-PC was therefore unable to stabilize the core structure but rather acted to destabilize it. In ref. 11, we developed a new trimer to address the problem. Another solution is to add more Ca²⁺ to the injection solution to induce a stronger aggregation in the core, as we did for the tail fin in the zebrafish. In the current study, we have performed a quick electrofunctionalization immediately after injection by applying 1.2 V for 2.5 min to stabilize the core structure and then allowed unpolymerized ETE-PC to diffuse radially, followed by a subsequent electrofunctionalization after 5 min to extend the dendrites in the tissue, which is mentioned in the methods.

The statement in manuscript “they also showed that mixtures of different concentrations of ETE derivatives affected the oxidation potential” indicates that mixing two different trimers,

ETE-PC and ETE-S, resulted in a gradually increasing oxidation potential based on the trimer ratio. In our manuscript, we mixed ETE-PC with another related compound, A5, which might affect the onset of oxidation. However, since our observed peak is at the expected onset of oxidation (0.47V), we do acknowledge that the discussion about possibly shifting oxidation potentials rather adds to confuse the reader. We have now changed the manuscript accordingly (line 171).

However, as per reviewer's suggestion, we have performed an additional experiment where we show how dendrite formation is affected by different A5:ETE-PC ratios. A5:ETE-PC solution was prepared by mixing A5 (20 mg ml⁻¹) with increasing concentrations of ETE-PC (10-50 mg ml⁻¹) and injected in to agarose gel followed by electrofunctionalization at 1 V for 5 min. As shown in the figure below, the dendrite formation improves as the concentration of ETE-PC increased and at a concentration of 40 mg ml⁻¹ or above well-defined dendrites were observed. **New data has been added as supplementary figure 2d in the revised manuscript.**

Fig. R1 (fig. S2d in revised manuscript): Effect of ETE-PC concentration on dendrite formation. A5:ETE-PC solution was prepared by mixing A5 (20 mg ml⁻¹) with increasing concentrations of ETE-PC (10-50 mg ml⁻¹) and injected in to agarose gel (0.5% in Ringer's solution) followed by electrofunctionalization at 1 V for 5 min.

2. Validation of the hydrogel property of A5:ETE-PC: The authors deliver the idea of hydrogel both to the in-situ electrofunctionalized core and dendrite through electrofunctionalization, as the chemical precursors were delivered as a water-soluble mixture. However, for precision on the terminology, we recommend the authors revisit the hydrogel properties of A5:ETE-PC. In general, hydrogels consist of a large amount of water content within its matrix innately, and the water content could hinder or enhance the characteristics or the performance of the material. We recommend the authors investigate the water content of both A5:ETE-PC core and dendrite, while comparing the impact of water content on its mechanical and electrical properties, as the material is naturally surrounded by a biofluid. Moreover, as most of injectable hydrogels do, the zwitterionic nature of the polymer network of A5:ETE-PC could suffer from excessive swelling in longer-term applications, while

also under its gradual biodegradation; thus, unless the electrophoretic therapy could completely eliminate the tumor immediately only with a single-term usage, it is inevitable that the performance should be prone to two significant properties that would affect simultaneously.

Response:

Yes, the reviewer is correct. We have thoroughly characterized A5 in ref 9. <https://doi.org/10.1021/acs.chemmater.1c04342> including the water content previously described. From the paper: “To further investigate these swelling characteristics, A5 solution (10 mg/mL) was added to isopropanol, which led to the formation of wires (see the Supporting Information). Drying of the wet wire to a dry state resulted in a 25× reduction of the cross-sectional area (see Figure S18). The addition of Ringer solution to the dry wire resulted in a 10-fold increase in the cross-sectional area, forming a hydrogel. With pure water, the structure was slowly dissolved. This shows that A5 form hydrogels with >90% water content and that ions are critical for maintaining the hydrogel structure.” The underlined part is important for the reviewer’s next question.

For A5 mixed with ETE-PC, we generated a “bulk hydrogel” by electrofunctionalizing A5:ETE-PC on an Au-coated glass slide. The gel was weighed before and after drying at 100°C. We measured a gel weight of 173 mg and a dry weight of 8 mg, resulting in a water content that is more than 20 times the polymer weight (A5 together with functionalized ETE-PC), thereby confirming the hydrogel nature of A5:ETE-PC. Unfortunately, the dendrites cannot be isolated from the A5:ETE-PC for additional investigations. **New data have been added as supplementary figure 2e in the revised manuscript.**

Figure R2. Bulk A5:ETE-PC hydrogel on an Au slide for estimating water content. The Au slide with the wet gel is tilted about 50 degrees without any noticeable gel slipping. A5:ETE-PC [20:40] was added onto an Au slide, and mixed 1:1 with Ringer’s solution, followed by electrofunctionalization where the Au slide acted as the anode. After electrofunctionalization, any remaining liquid was removed by careful pipetting and the slide was weighed. The gel was dried on a hot plate at 100 °C, followed by weighing.

3. Bioresorbable property of A5:ETE-PC hydrogel: The claim that the ETE-PC hydrogel is bioresorbable lacks adequate experimental data. Both A5 and ETE-PC contain thiophene rings that are generally considered not to be hydrolysable, raising doubts about their resorbability. Although reference 10 claims that A5 may exhibit superior bioresorption than PEDOT: PSS, this comparison is inadequate, as PEDOT:PSS is widely recognized as non-

bioresorbable. Referencing prior work that does not directly address the degradation kinetics of A5, or ETE-PC is insufficient to support the bioresorbable nature of the material. To claim the biocompatibility of the bioresorption of A5:ETE-PC, we recommend the authors to further investigate the biocompatibility of the degradation byproduct, especially regarding the ability of its compatibility with the natural metabolic mechanism.

Response: We appreciate the reviewer's critical observation on bioresorption of A5:ETE-PC. This manuscript does not specifically address bioresorption; instead, we refer to our earlier publications, which demonstrate that the polymer completely disappears from the heart, brain, and tail fin of zebrafish. Although the tail fin is an outlier, the brain and heart tissue are expected to be conserved across species. Our previous studies also show that, following an initial inflammation after the injection, the inflammation resolves. Thus, there appears to be no ongoing activity of macrophages/microglia in the brain.

Unlike most other polymers, the A5 is small with an average length of 7-8 monomers, and a maximum of 12. The small size makes them amenable to renal clearance. Hence, small oligomers of eroded A5:ETE-PC are likely to be cleared through the renal route.

We believe the clearance results from bioerosion of highly soluble oligomer strands. To verify this as a plausible mechanism, we performed an in vitro experiment in which we prepared an A5:ETE-PC electrode on an agarose gel and maintained it in PBS (pH 7.4) at 37 °C for up to two weeks. As shown in Figure R3, A5:ETE-PC with no electrofunctionalization has disappeared within 7 days. Furthermore, the electrofunctionalized A5:ETE-PC was also found to erode over time; however, the erosion rate depends on the duration of electrofunctionalization (**data added as Supplementary Figure S5 in the revised manuscript**). At longer electrofunctionalization times, erosion is slower; thus, this has the potential to predictably modulate the bioresorption as per the therapeutic window (for short to long term use).

Figure R3. Erosion of A5:ETE-PC in physiological buffer (PBS). A5:ETE-PC (20:40 mg ml⁻¹) was added on agarose gel (0.5% in Ringer's solution) and left to diffuse for 2 min, followed by electrofunctionalization (EF) for various durations (0–5 min). After the EF, the agarose

blocks with A5:ETE-PC were submerged in PBS (pH 7.4) and kept on mild shaking at 37 °C. The erosion was monitored by imaging at various time points (0–14 days).

We also evaluated a complementary mechanism. In vitro assays demonstrate that macrophages phagocytose A5:ETE-PC, providing a plausible uptake route if larger material fragments are shed after IRE treatment (**Supplementary Fig. S6 in the revised manuscript**). Notably, in line with findings reported by Wei Gao et al., we observe an accumulation of immune cells around the electrode following IRE (Fig. 4c).

To put in context:

Yes, PEDOT:PSS is metabolically stable; additionally, PEDOT itself is insoluble in both water and organic solvents. Two recent publications on bioresorption have focused on replacing the PSS in PEDOT:PSS with glycosides to make it more biodegradable. Wei Gao, in Nature Communications <https://doi.org/10.1038/s41467-025-59045-1>, used enzymes to degrade the glycosidic part; however, the insoluble PEDOT remained intact, and the fate of this material is unclear, as illustrated in the supplementary figure [Fig S9], there is remaining polymer in the bioresorption study even after 11 weeks. In contrast, we see total disappearance of A5:ETE-Rs in our previous studies. Wei Gao potentially alludes to the degradation occurring in cells from the immune system, as he observes an increase in them around the polymer. Molly Stevens and coworkers achieve this more elegantly by covalently linking the glycoside to the PEDOT in Advanced Healthcare Materials <https://doi.org/10.1002/adhm.202403995>. Thus, after enzymatic degradation of the glycoside, the intact PEDOT is attached to the glycoside fragments, making the material soluble and bioerodible.

We designed a highly soluble PEDOT-S derivative (A5) that aggregates in tissue because it incorporates ions, thoroughly discussed in ref 9. Thus, we showed (see above) that when ions (predominantly Ca²⁺) are depleted from the conductive structure, small oligomers (Mw 600-4500 Da) are gradually released. These *hydrophilic and anionic/zwitterionic* oligomers are expected to have a low affinity for protein binding leading to a rapid distribution in plasma & interstitial fluid. Their small hydrodynamic size ≈2–3 nm, allows them to be freely filtered by the kidneys. Thus, we do not anticipate any metabolism, and we have not seen indications of toxicity in vivo.

In addition, we <https://doi.org/10.1002/adv.202408628> and others <https://doi.org/10.1021/ja045835e> have shown that polymerizing thiophene trimers yields the same sizes as PEDOT derivatives, suggesting that there may be a size limit for EDOT-based polymers.

Gordon Wallace also demonstrated this by using PEDOT-COOH instead of PEDOT-Sulfonate (A5). <https://doi.org/10.1039/D2SC06342E>. Although PEDOT-COOH is not very soluble, it completely disappears in vivo. We have adopted their in vitro studies to show that bioerosion is a viable mechanism, and they show the same results.

In summary, both our PEDOT-derivative, A5, and the ETE-PC are highly soluble and come at a small size, making them amenable to renal clearance when not aggregated. Larger pieces may

also be cleared by the immune system, as observed in macrophages that are prone to engulfing the A5:ETE-PC.

4. Biocompatibility of A5:ETE-PC hydrogel: The authors state that both A5 and ETE-PC are biocompatible (page 13, line 380-382). However, if the hydrogel is indeed bioresorbable as claimed, its biocompatibility must also account for the concentration and nature of its degradation byproducts. Toxicity is often dose-dependent and can arise from accumulated degradation products, even if the parent materials are considered biocompatible. Therefore, additional experimental data are needed to assess how varying concentrations of A5 affect cell viability or tissue response.

Response: We acknowledge the reviewer's concern; however, in our previous study, we demonstrated the effect of varying concentrations of A5 on normal lung fibroblast cells, where we observed no toxic effect of A5 on cell viability at a dose as high as 1 mg ml⁻¹ (<https://doi.org/10.1038/s41467-023-40175-3>). In the same study, we investigated the toxicity of ETE-PC and observed only a minor effect at the highest dose (1 mg ml⁻¹). And as discussed above, we do not expect any degradation byproducts since the A5 oligomers are already at a size suitable for renal clearance.

5. Irreversible electroporation (IRE) efficacy: The authors claim that IRE using the hydrogel achieves effective tumor ablation, enhanced by immunogenic cell death and increased cell infiltration. However, the manuscript lacks direct evidence demonstrating the effect of IRE on normal cells. To validate the therapeutic efficacy of the IRE treatment, an additional control group – consisting of normal cells treated with and without IRE-should be included.

Response: IRE may affect both normal and cancer cells, and it is known that tumor cells have a higher susceptibility towards IRE compared to other cells (<https://doi.org/10.1038/srep17157>). However, assembly of the electrode within the tumor, where only tumor tissue is exposed to the electrode, makes it a tumor-specific therapy. Furthermore, the non-thermal nature of IRE treatment minimizes the risk of affecting nearby normal tissue. (<https://doi.org/10.1016/j.jvir.2014.01.028>)

6. Homogeneity of dendrite structure: The authors claim that the ETE-PC dendrite structure shows 'uniform conductivity' throughout the entire network, as all chemical compositions (A5, ETE-PC) are intrinsically conductive (page 9, line 261). However, this simple inference does not validate reality, as there still exists possibility of local structure formations that could hinder the uniformity of conductivity, such as local electrostatic aggregation, microphase separation, etc. Regarding the issue, we recommend the authors provide detailed information on the chemical and physical properties of the in-situ formed dendrite microstructures, compared to the 'core' of A5:ETE-PC hydrogel, especially regarding the chemical homogeneity, mechanical modulus and adhesiveness. Although dendrite itself holds the core significance of the following findings, there exists limited evidence regarding the issue.

Response: We thank the reviewer for this critical comment. Here, we aim to establish a contrast between our structure, which is entirely composed of intrinsically conductive materials, and other hydrogels that utilize conductive fillers to enhance their electrical properties. This makes our electrode conductive throughout the entire network. However, we understand that the phrase “uniformly conductive” may create the impression that the structure has a similar extent of conductivity throughout the entire network. We don’t claim that. We have edited the concerning part in the revised manuscript.

Related to the discussion about local aggregation, microphase separation, etc.: During the electrofunctionalization step, core regions with higher resistivity (as could be imagined arising from phase separation) would also experience the highest potential drop and thus be a preferential site for ETE-PC attachment. The electrofunctionalization will therefore not only create dendrites extending radially, but it will also act to “heal” resistive regions in the core, making it electrically more homogeneous over its entire length.

We have conducted additional experiments to investigate the mechanical properties of A5:ETE-PC in tumor cavities. Similar to what we have observed for agarose and zebrafish brains and hearts, the polymer adapts to the surrounding material. In the cavities, we find that both pristine cavities (without polymer) and polymer-coated cavities have low Young’s moduli of 1–4kPa. This data has been added to the supplemental file (supplementary figure S16 in the revised manuscript).

7. Control experiments on electrical characterization: The authors repeatedly claim that A5:ETE-PC shows superior electrical properties by its enhanced interfacing (increase in surface area due to dendrite formation), referring to its decrease in impedance compared to mere A5 or metal electrodes (lines 192 (page 6), 215 (page 7), figures S3(e), and S7(b)). However, to highlight the sole effect of the dendrite structure on impedance, the authors should compare the provided data with a bulk A5:ETE-PC hydrogel. Assuming there does exist a strong and stable interface between A5:ETE-PC dendrite or bulk and tumor area, a controlled ex-vivo impedance analysis would be a suggestable setup for further investigation.

Response: To highlight the effect of dendrites on impedance, we have performed impedance analysis before and after electrofunctionalization, i.e., before and after dendrite formation. The results show that the A5:ETE-PC before electrofunctionalization demonstrates a decrease in impedance, which is further decreased significantly post-electrofunctionalization as compared to the metal electrode. This suggests that the bulk itself, A5:ETE-PC, exhibits good conductive property; however, electrofunctionalization significantly enhances it, suggesting the additive effect of dendrite formation. **The data has been included in the revised manuscript as supplementary figure S10c.**

Figure R4. EIS characterization was performed to examine impedance in the tumor using metal (Pt-Pd coated capillary) or A5:ETE-PC electrode before and after electrofunctionalization.

8. Adhesion and mechanical stability of A5:ETE-PC: The authors claim that the dendrite structure forms enrichment in interfacing, resulting in lower impedance and enhanced electrophoresis efficacy. However, no evidence exists on the mechanical stability of the dendrite – tumor tissue interface, which in turn could critically impact the actual IRE therapy. Provided in the prior studies with similar chemical compositions (reference 10, 11), the shear moduli of are extremely low, initially to enable the mechanical match with benign brain tissue. However, such low moduli could further act as the bottleneck on stable interfacing in vivo. Authors should provide detailed information about the stability of interfacing of dendrites and tumor cells, using indentation methods.

Response: We have now performed mechanical measurements in indentation mode, as the reviewer suggested. As in our previous studies, we find that the polymer adapts to the surrounding tissue, exhibiting Young's moduli of 1–4kPa. We view mechanical matching as a benefit, enabling the tissue and polymer to respond similarly to external stresses. During drape formation, the polymer shows strong interfacial adhesion to the cavity wall: when the bulk polymer is withdrawn, a thin drape remains lining the cavity. This drape withstands extensive washing (e.g., multiple rinses and aspirations) and remains intact. Because these steps did not alter outcomes, we omitted them from the formal procedure to keep it concise. Hence, after electrofunctionalization, our drape model demonstrates a uniform layer of A5:ETE-PC strongly adhered to the tissue in the cavity lining with dendrites extending into the tissue (main figure 2d & e). Moreover, when the cavity was flushed to remove the electrofunctionalized A5:ETE-PC layer, the layer remained intact, indicating strong bioadhesivity.

9. Time-resolved analysis of electrofunctionalization: The authors only visualizes the already-formed A5:ETE-PC after the electrofunctionalization (page 7, lines 212-214, figure S5). To further ensure that the polymerization step takes a short amount of time that minimally affects the physiological activity of the surrounding benign tissues, we recommend the authors provide the time-resolved visualization of electrofunctionalization in situ. Additionally, the provided visualization (figure S5, S6) does not precisely mimic the actual environment of electrofunctionalization, as here the process occurs on the scaffold of A5, rather than the injected solution of precursors and its core. Moreover, authors should provide detailed information on the electrofunctionalization process, as there exists no clear elaboration on why the process is divided into two steps (step 1: 1.2V for 2.5 minutes, step 2: 3V for 10 minutes).

Response: As explained in response to comment 1, an initial electrofunctionalization (1.2V for 2.5 minutes) was performed to stabilize A5:ETE-PC core, followed by a second one (3V for 10 minutes) to extend the dendrites. We now discuss this in the revised manuscript. The long-duration electrofunctionalization was performed to ensure the formation of elongated, dense dendrites for better therapeutic applications. Since the whole process happens in tumor tissue, it minimally affects the physiological activity of the surrounding benign tissues. In fact, another form of cancer electrotherapy (electrochemical therapy) employs low voltages (up to 12 V) for long duration (1.5 – 4 hrs), which is generally considered safe, with low risk of complications.

10. (Page 9, lines 267–269) Please explain the terminology EDD, as it initially uses the term here.

Response: EDD stands for embryonic development day of chicken embryo. We get fertilized eggs which do not start developing until kept in specialized incubator with optimum conditions (temperature and humidity). When eggs are placed in the incubator is denoted as embryonic development day 0 (EDD 0). As per reviewer's suggestion the terminology is explained under the heading "Chicken chorioallantoic membrane (CAM) tumor model of glioblastoma" (page 7, line 203-205), where the term is used first, in the revised manuscript.

11. (Page 13, line 360) Please reassign the figure that matches the explanation "Notably, 800 V cm⁻¹ and 1000 V cm⁻¹ pulses 360 caused a complete loss of U87 cell viability", rather than figure 1h.

Response: We thank the reviewer for highlighting this error. The correction has been made in the revised manuscript.

Reviewer #2 (Remarks to the Author):

In this paper, the authors developed a new electroimmunotherapy for glioblastoma using an injectable conductive hydrogel. Combination of the injectable conductive hydrogel and the irreversible electroporation (IRE) method significantly reduces the invasiveness of the patients compared to conventional treatments. Moreover, when combined with surgery, it is also effective in eradicating remaining tumor cells, suggesting the potential for becoming a groundbreaking treatment. The same material (A5: ETE-PC) as in the previous study (reference 10) was used, and although there is little novelty in the material, the validity and effectiveness of the evaluation using a chicken chorioallantoic membrane (CAM) tumor model (glioma) are demonstrated. However, several parts are difficult for readers outside the field to understand, and I am convinced that the value of this paper will be further increased by revising it with the following points in mind.

1) There is a lack of information on how the material used (A5: ETE-PC) was optimized. It would be desirable to supplement the process by which the structure, molecular weight, concentration, and A5/ETE-PC mixing ratio were determined to be used.

Response: The synthesis and characterization of A5 and ETE-PC has been shown in our previous articles (Chemistry of Materials. 2022;34(6):2752-63, *Adv. Funct. Mater.* **2022**, 2202292). The concentration and mixing ratio were determined based on ease of working with the material. After trying various mixing ratios, we found that 40 mg/ml of ETE-PC and 20 mg/ml of A5 are the most suitable concentrations.

However, as per reviewer's suggestion, we have performed an additional experiment where we show how electrode formation is affected by different A5:ETE-PC ratios. A5:ETE-PC solution was prepared by mixing A5 (20 mg ml⁻¹) with increasing concentrations of ETE-PC (10-50 mg ml⁻¹) and injected into agarose gel followed by electrofunctionalization at 1 V for 5 min. As shown in the figure below, the dendrite formation improves as the concentration of ETE-PC increased and at a concentration of 40 mg ml⁻¹ or above well-defined dendrites were observed. New data has been added as supplementary figure S2d in the revised manuscript.

Fig. R1 (fig. S2d in revised manuscript): Effect of ETE-PC concentration on dendrite formation. A5:ETE-PC solution was prepared by mixing A5 (20 mg ml⁻¹) with increasing concentrations of ETE-PC (10-50 mg ml⁻¹) and injected in to agarose gel (0.5% in Ringer's solution) followed by electrofunctionalization at 1 V for 5 min.

2) Considering the molecular structure, it is unlikely that the polymer used will be degraded in the body. I presume that the term "bioabsorbable" refers to the disappearance of this polymer from the site of injection into the body. However, information on the pharmacokinetics of these polymers after injection is essential. Information on the organ distribution and excretion from the body after administration into the mammalian brain is required.

Response: We appreciate the reviewer's critical observation on bioresorption of A5:ETE-PC. This manuscript does not specifically address bioresorption; instead, we refer to our earlier publications, which demonstrate that the polymer completely disappears from the heart, brain, and tail fin of zebrafish. Although the tail fin is an outlier, the brain and heart tissue are expected to be conserved across species. Our previous studies also show that, following an initial inflammation after the injection, the inflammation resolves. Thus, there appears to be no ongoing activity of macrophages/microglia in the brain.

Unlike most other polymers, the A5 is small with an average length of 7-8 monomers, and a maximum of 12. The small size makes them amenable to renal clearance. Hence, small oligomers of eroded A5:ETE-PC are likely to be cleared through the renal route.

We believe the clearance results from bioerosion of highly soluble oligomer strands. To verify this as a plausible mechanism, we performed an in vitro experiment in which we prepared an A5:ETE-PC electrode on an agarose gel and maintained it in PBS (pH 7.4) at 37 °C for up to two weeks. As shown in Figure R3, A5:ETE-PC with no electrofunctionalization has disappeared within 7 days. Furthermore, the electrofunctionalized A5:ETE-PC was also found to erode over time; however, the erosion rate depends on the duration of electrofunctionalization (data added as Supplementary Figure S5 in the revised manuscript). At longer electrofunctionalization times, erosion is slower; thus, this has the potential to predictably modulate the bioresorption as per the therapeutic window (for short to long term use).

We also evaluated a complementary mechanism. In vitro assays demonstrate that macrophages phagocytose A5:ETE-PC, providing a plausible uptake route if larger material fragments are shed after IRE treatment (Supplementary Fig. S6 in the revised manuscript). Notably, in line with findings reported by Wei Gao et al., we observe an accumulation of immune cells around the electrode following IRE (Fig. 4c).

Figure R3. Erosion of A5:ETE-PC in physiological buffer (PBS). A5:ETE-PC (20:40 mg ml⁻¹) was added on agarose gel (0.5% in Ringer's solution) and left to diffuse for 2 min, followed by electrofunctionalization (EF) for various durations (0–5 min). After the EF, the agarose blocks with A5:ETE-PC were submerged in PBS (pH 7.4) and kept on mild shaking at 37 °C. The erosion was monitored by imaging at various time points (0–14 days).

To put in context:

Yes, PEDOT:PSS is metabolically stable; additionally, PEDOT itself is insoluble in both water and organic solvents. Two recent publications on bioresorption have focused on replacing the PSS in PEDOT:PSS with glycosides to make it more biodegradable. Wei Gao, in Nature Communications <https://doi.org/10.1038/s41467-025-59045-1>, used enzymes to degrade the glycosidic part; however, the insoluble PEDOT remained intact, and the fate of this material is unclear, as illustrated in the supplementary figure [Fig S9], there is remaining polymer in the bioresorption study even after 11 weeks. In contrast, we see total disappearance of A5:ETE-Rs in our previous studies. Wei Gao potentially alludes to the degradation occurring in cells from the immune system, as he observes an increase in them around the polymer. Molly Stevens and coworkers achieve this more elegantly by covalently linking the glycoside to the PEDOT in Advanced Healthcare Materials <https://doi.org/10.1002/adhm.202403995>. Thus, after enzymatic degradation of the glycoside, the intact PEDOT is attached to the glycoside fragments, making the material soluble and bioerodible.

We designed a highly soluble PEDOT-S derivative (A5) that aggregates in tissue because it incorporates ions, thoroughly discussed in ref 9. Thus, we showed (see above) that when ions (predominantly Ca²⁺) are depleted from the conductive structure, small oligomers (Mw 600-4500 Da) are gradually released. These *hydrophilic and anionic/zwitterionic* oligomers are expected to have a low affinity for protein binding leading to a rapid distribution in plasma & interstitial fluid. Their small hydrodynamic size ≈2–3 nm, allows them to be freely filtered by the kidneys. Thus, we do not anticipate any metabolism, and we have not seen indications of toxicity in vivo.

In addition, we <https://doi.org/10.1002/advs.202408628> and others <https://doi.org/10.1021/ja045835e> have shown that polymerizing thiophene trimers yields the same sizes as PEDOT derivatives, suggesting that there may be a size limit for EDOT-based polymers.

Gordon Wallace also demonstrated this by using PEDOT-COOH instead of PEDOT-Sulfonate (A5). <https://doi.org/10.1039/D2SC06342E>. Although PEDOT-COOH is not very soluble, it completely disappears in vivo. We have adopted their in vitro studies to show that bioerosion is a viable mechanism, and they show the same results.

In summary, both our PEDOT-derivative, A5, and the ETE-PC are highly soluble and come at a small size, making them amenable to renal clearance when not aggregated. Larger pieces may also be cleared by the immune system, as observed in macrophages that are prone to engulfing the A5:ETE-PC.

3) A voltage was applied, but a clear diagram of how this works is necessary. A clear diagram of the agarose system was attached; however, it would be even more beneficial if there were a diagram that clearly illustrates the relationship between the electrodes and the gel injection site in the CAM system.

Response: As per the reviewer's suggestion, a schematic representation of the in vivo electrode assembly and its use for IRE has been added in the revised manuscript as supplementary figure S26.

REVIEWER COMMENTS

We thank the reviewers for carefully reading through and helping us to improve the manuscript.

Reviewer #1 (Remarks to the Author):

I have some additional suggestions for perfecting this study with robust supporting data. Additional suggestions:

Comment: Although electrofunctionalization occurs within the area of tumor tissue, it is still vaguely described that only the tumor tissue is specifically targeted as the functionalization region and limited with its propagation or actual function on surrounding normal tissues, in a spatial-wise point of view. Although extremely difficult to formulate an in-situ imaging method for such, it would be much more impactful and intuitively persuasive in terms of the biocompatibility of the novel process. One simple suggestion is to perform histological analyses on both types of tissues after the treatment and show the compatibility of the treatment on the normal region.

Response: We thank the reviewer for the suggestion. We reiterate that IRE may affect both normal and cancer cells. In fact, various studies have assessed efficacy of IRE in tissue ablation in normal tissue like liver. So, it is not a question of tissue specificity but more of the location where the IRE is applied, and since in our case, assembly of the electrode is within the tumor, it mainly affects the tumor. Further, in the chicken chorioallantois membrane (CAM) model, the tumor grows on the surface of a very thin CAM (as shown in fig. R1A) and is not surrounded by any thick normal tissue to inject.

Figure. R1: Effect of IRE on CAM adjacent to tumor (A) U87 tumor on CAM (B) Harvested U87 tumor along with CAM (C) Bright field and DAPI stained images of tumor + CAM cryosections at 4x and 10 x magnifications.

To show the effect of IRE on nearby normal tissue, we injected A5:ETE-PC into the tumor and applied IRE. We harvested the tumor along with CAM (as nearby normal tissue) attached to the tumor, as shown in figure R1B and prepared the sections. DAPI staining is used to identify surviving cells in the treated tissue, with DAPI binding to AT-rich DNA regions and emitting blue fluorescence under a microscope. This helps differentiate ablated areas from viable cells that have not been sufficiently

affected by the electrical pulses, which allows for assessing the effectiveness of the IRE treatment and the presence of any surviving or partially ablated tissue. Areas with no nuclei or compromised nuclei indicate successful IRE-induced cell death. As shown in figure R1C, negligible DAPI staining is observed in most of the tumor. In contrast, distinct nuclear staining is present in the CAM part of the section, suggesting that IRE mainly affected the tumor, but not the CAM. It is an indication that the assembly of electrodes within the tumor is unlikely to affect nearby normal tissue post IRE.

Comment: Although repeated claims about the existence of reports on the Young’s modulus of A5:ETE-PC as 1-4 kPa, there consistently seems no specific data shown in this article and the referenced paper (Hjort, M., Mousa, A.H., Bliman, D. et al. In situ assembly of bioresorbable organic bioelectronics in the brain. Nat Commun 14, 4453 (2023). I strongly recommend the authors show the direct measurements of Young’s modulus, with results from indentations. It would much more easily support the claim with the benefit of low mechanical stiffness to adhesive interfacing with tumor tissues, when the actual Young’s modulus data and relationship with adhesion is displayed in parallel, with appropriate control group.

Response: We have now added the initial indentation curves together with fits for a reference resection and two drape-covered tumor resections to the supplemental information (Fig. 16). Source data for the force curve and Hertz equation fits have been added to the source data file.

The Biomomentum indentation fits the Hertzian approximation for a spherical indenter. More specifically, the following is fitted:

$$P = \frac{4\sqrt{R}}{3} \frac{E}{(1 - \nu^2)} h^{3/2} = \frac{8\sqrt{R}}{3} [2G]h^{3/2}$$

With P as the force, R the indenter radius, h is the displacement, E the elastic modulus, nu the Poisson ratio (for soft tissue often taken between 0.45 and 0.5), G the shear modulus.

Fig R2: Indentation measurements to deduce Young’s modulus. Initial indentation of a pristine tumor cavity sample (reference) and two drape covered tumor resections (tumor 1/2). The red lines show the Hertzian fit as given by the Biomomentum software. Please note that the curves do not return to 0 force, suggesting adhesion. Young’s moduli were estimated assuming about 7mm thickness and 0.45 for the Poisson ratio.

Reviewer #2 (Remarks to the Author):

In the revised paper, the authors developed a new electroimmunotherapy for glioblastoma using an injectable conductive hydrogel. Combination of the injectable conductive hydrogel and the irreversible electroporation (IRE) method significantly reduces the invasiveness of the patients compared to conventional treatments. Moreover, when combined with surgery, it is also effective in eradicating remaining tumor cells, suggesting the potential for becoming a groundbreaking treatment. The same material (A5: ETE-PC) as in the previous study (reference 10) was used, and although there is little novelty in the material, the validity and effectiveness of the evaluation using a chicken chorioallantoic membrane (CAM) tumor model (glioma) are demonstrated.

The authors have responded sincerely and fully to all reviewer comments, and the content and value of this paper have been significantly improved. Therefore, I have determined that the paper can be accepted as is.

Response: We are thankful to the reviewer for recommending our manuscript for publication and for providing insightful comments to improve the study.

REVIEWER COMMENTS

We thank the reviewers for carefully reading through and helping us to improve the manuscript.

Reviewer #1 (Remarks to the Author):

I think now the manuscript is well prepared for the publication with robust explanation for reviewers' comments.

Response: We are thankful to the reviewer for recommending our manuscript for publication and for his critical review of the manuscript to improve the study.